# Dynamical SPQEIR model assesses the effectiveness of non-pharmaceutical interventions against COVID-19 epidemic outbreaks

Daniele Proverbio[1]*, Françoise Kemp[1], Stefano Magni[1], Andreas Husch[1], Atte Aalto[1], Laurent Mombaerts[1], Alexander Skupin[1], Jorge Gonçalves[1], Jose Ameijeiras-Alonso[2], Christophe Ley[3]

1 Luxembourg Centre for Systems Biomedicine, University of Luxembourg, Belvaux, Luxembourg,
2 Department of Mathematics, U Leuven, Leuven, Belgium, 3 Department of Applied Mathematics, Computer Science and Statistics, Ghent University, Ghent, Belgium

* daniele.proverbio@uni.lu

**Data Availability Statement:** Databases of social measures can be accessed at https://www.who.int/emergencies/diseases/novel-coronavirus-2019/phsm. ACAPS database is at https://www.acaps.org/covid19-government-measures-dataset.

## Abstract

Against the current COVID-19 pandemic, governments worldwide have devised a variety of non-pharmaceutical interventions to mitigate it. However, it is generally difficult to estimate the joint impact of different control strategies. In this paper, we tackle this question with an extended epidemic SEIR model, informed by a socio-political classification of different interventions. First, we inquire the conceptual effect of mitigation parameters on the infection curve. Then, we illustrate the potential of our model to reproduce and explain empirical data from a number of countries, to perform cross-country comparisons. This gives information on the best synergies of interventions to control epidemic outbreaks while minimising impact on socio-economic needs. For instance, our results suggest that, while rapid and strong lockdown is an effective pandemic mitigation measure, a combination of social distancing and early contact tracing can achieve similar mitigation synergistically, while keeping lower isolation rates. This quantitative understanding can support the establishment of mid- and long-term interventions, to prepare containment strategies against further outbreaks. This paper also provides an online tool that allows researchers and decision makers to interactively simulate diverse scenarios with our model.

## Introduction

The current global COVID-19 epidemic has led to significant impairments of public life world-wide. To mitigate the spread of the virus and to prevent dramatic situations in the healthcare systems, many countries have implemented a combination of rigorous measures like lockdown, isolation of symptomatic cases and the tracing, testing, and quarantine of their contacts. In order to obtain information about the efficacy of such measures, a quantitative understanding of their impact is necessary. This can be based on statistical methods [1] and on

Worldwide epidemiological data collection from John Hopkins University is at https://github.com/CSSEGISandData/COVID-19. Lombardy data were retrived from https://github.com/pcm-dpc/COVID-19. Google mobility data were accessed through https://ourworldindata.org/covid-mobility-trends. The code for analysis can be found at https://github.com/daniele-proverbio/assessing_strategies.

**Funding:** DP and SM's work is supported by the FNR PRIDE DTU CriTiCS, ref 10907093. FK's work is supported by the Luxembourg National Research Fund PRIDE17/12244779/PARK-QC. A. H. work was partially supported by the Fondation Cancer Luxembourg. JG is partly supported by the 111 Project on Computational Intelligence and Intelligent Control, ref B18024. AA is supported by the Luxembourg National Research Fund (FNR) (Project code: 13684479). JAA is supported by the FWO research project G.0826.15N (Flemish Science Foundation), GOA/12/014 project (Research Fund KU Leuven), Project MTM2016-76969-P from the Spanish State Research Agency (AEI) co–funded by the European Regional Development Fund (ERDF) and the Competitive Reference Groups 2017–2020 (ED431C 2017/38) from the Xunta de Galicia through the ERDF.

**Competing interests:** The authors declare no competing interests.

epidemiological models [2]. Epidemiological modeling in particular can provide detailed mechanisms for the epidemic dynamics and allow investigating how epidemics will develop under different assumptions.

Preliminary efforts have been made to quantify the contribution of different policy interventions [3], but these rely on complex models based on a number of assumptions. Instead, we base our study on a classical SEIR-like epidemiological model. SEIR models are minimal mechanistic models that consider individuals transitioning through Susceptible → Exposed → Infectious → Removed state during the epidemics [4]. The essential control parameter is the basic reproduction number $R_0$ [5], that worldwide non-pharmaceutical mitigation strategies aim at reducing below the threshold value 1. Several literature studies consider the effect of single interventions in SEIR-like models [6–8]. We aim at considering the added value of early interventions, namely those that target Susceptible and Exposed people, and the effect of different combinations of control strategies on the infection curve. To do so, we incorporate additional compartments reflecting different categories of control strategies, identified by socio-political studies [9]. In particular, the model focuses on four main mitigation programs: social distancing (lowering the rate of social contacts), active protection (decreasing the number of susceptible people), active removal of latent asymptomatic carriers [10], and active removal of infectious carriers. This study investigates how these programs achieve mitigation both individually and combined, first conceptually and then by cross-country comparison. By our modelling choice, we consider how and how much preventive interventions can supplement the quarantining of contagious individuals. We ultimately show that analogous containment levels of the infectious curve can be achieved by alternative synergies of non-pharmaceutical interventions. This information can supply Government decisions, helping to avoid overloading the healthcare system and to minimise stressing the economic system (due to prolonged lockdown). We expect our model, together with its interactive online tool, to contribute to crucial tasks of decision making and to prepare containment strategies against further outbreaks.

## Materials and methods

This study links policy measures to epidemiological modelling, focusing on how the dynamics of the infectious curve is controlled by several interventions. Initially, we perform a conceptual analysis, like in other works [11, 12]. Then, we investigate how well the considered control synergies reproduce and explain the evolution of empirical data from the first COVID-19 wave in six different countries. By doing so, we hope to contribute to discussions about the relevance of such conceptual strategies in real-world conditions. In this section, we illustrate the modelling choices and the use of data.

### The classical SEIR model

SEIR models are continuous-time, mass conservative compartment-based models of infectious diseases [4, 13]. They assume a homogeneously mixing population (or fully connected graphs) and focus on the evolution of mean properties of the closed system. All of these models are classical and widely used tools to investigate the principal mechanisms governing the spread of infections and their dynamics. There is a broad range of such models, from more conceptual to more realistic versions, e.g. SEIR with delay [14], spatial coupling [15, 16], extended compartments [17], or those that consider progression of treatments and age distribution [18].

Main compartments of SEIR models (see Fig 1, framed) are: susceptible S (the pool of individuals socially active and at risk of infection), exposed E (corresponding to latent carriers of the infection), infectious I (individuals having developed the disease and being contagious) and removed R (those that have processed the disease, being either recovered or dead). The

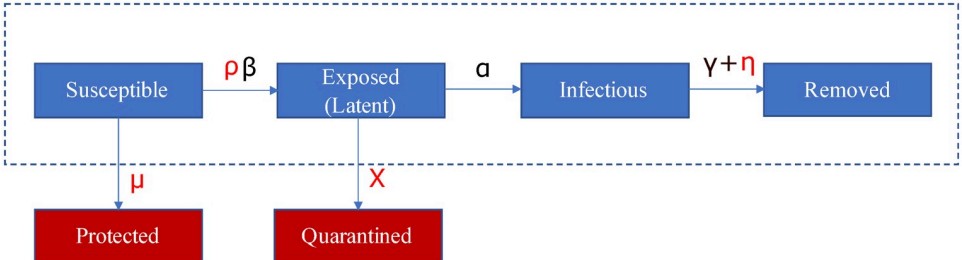

**Fig 1. Scheme of the SPQEIR model.** The basic SEIR model (framed blue blocks) is extended by the red blocks to the SPQEIR model. Parameters that are linked to mitigation strategies are shown in red. Interpretation and values of parameters are given in Table 2.

model's default parameters are the average contact rate $\beta$, the inverse of mean incubation period $\alpha$ and the inverse of mean contagious period $\gamma$. When focusing on infection dynamics rather than patients' fate, the latter combines recovery and death rate [19]. From these parameters, epidemiologists calculate the "basic reproduction number" $R_0 = \beta/\gamma$ [20] at the epidemic beginning. During the epidemic progression, isolation after diagnosis, vaccination campaigns and active mitigation measures are in action. Hence, we speak of "effective reproduction number" $\hat{R}(T)$ [21].

## Data and analyzed countries

When investigating the ability of our conceptual model to explain mitigation, we compared it with empirical data. To do this, we considered the main non-pharmaceutical interventions applied by several countries, by integrating multi-disciplinary information. In fact, governments worldwide have issued a number of social measures, including those for public health safeguard, economic support, movement restriction and non-pharmaceutical interventions to hamper disease spreading. Scholars from political sciences and sociology have recorded and classified such measures [22, 23]. Among the resources listed on the World Health Organization "Tracking Public Health and Policy Measures" [9], we used information from the ACAPS database [24] that contains a curated categorization of policy measures. ACAPS is an independent, non-profit information provider helping humanitarian actors to respond more effectively to disasters. The ACAPS analysis team has aggregated and classified interventions from different sources (media, governments and international organizations), for all countries and in time. Mitigation measures against the epidemic are classified under "Movement restrictions", "Lockdown", "Social Distancing" and "Monitoring and Surveillance". Our modelling choice is based on these categories, which are reflected by additional compartments to the classical SEIR model (see next section).

Epidemiological data for all selected countries and regions were obtained from the COVID-19 Data Repository by the Center for Systems Science and Engineering (CSSE) at Johns Hopkins University [25]. The data are from 22 Jan 2020 to 08 July 2020. Lombardy data were obtained from the Protezione Civile Italiana data repository "Dati COVID-19 Italia" [26], from 22 Feb 2020 to 08 July 2020. We acknowledge that the quality of data of early detection of COVID-19 cases is often associated with limited testing capabilities, which could bias subsequent analysis. However, across the analysed countries, the share of positive tests was similar (see e.g. [27]), and no significant deviation from expected dynamics was observed by studies applying Benford's law [28, 29], possibly indicating that these data still capture to a reasonable extent the dynamics of the epidemic wave. In addition, the analysed countries were selected

based on the fact that they sufficiently met the other model assumptions, e.g. low spatial heterogeneity and large amount of cases to fulfill the mean field assumption.

This study analyses the effect of mitigation measures in flattening the curve. Despite having a precise starting date, such measures take some days to be fully effective. We estimate an average delay using the Google Mobility Reports [27, 30] for the selected countries. Google provides changes in mobility with respect to a monthly baseline, w.r.t. 6 locations: Retail & Recreation, Grocery & Pharmacy, Transit stations, Workplaces, Residential, Parks. We average the decrease in mobility at the first four locations (corresponding to those where social mixing happens more frequently [31]) to get a proxy of the time needed for hard lockdown to be fully effective (cf. Fig 4c).

## The extended SPQEIR model to reflect suppression strategies

SEIR models reproduce the typical bell-shaped epidemic curves for the number of infected people. The dynamics of this curve is of high importance for practical policy making. Not only it relates to the main stressors for the health system [17, 18, 31], but it also has an impact on the economic system [32–34], e.g. because it takes some time ($\mathbb{T}$) to mitigate the curve, until the number of new infections is below an accepted threshold. Commonly, mitigation measures against epidemics aim at flattening the curve of new infections [10]. However, the classical SEIR model is not granular enough to investigate mitigation measures when they need to be considered or should be sequentially reduced if already in place. Therefore, we extend the classical SEIR model as in Fig 1 (red insertions) into the SPQEIR model, to reflect the intervention categories described above. We particularly focus on the control of Susceptible and Exposed people, given by preventive isolation, contact tracing or social distancing measures, but we also include the control of infectious people by isolation. The model can be summarized as follows:

- The classical blocks S, E, I, R are maintained;

- A social distancing parameter $\rho$ is included to tune the contact rate $\beta$;

- Two new compartments are introduced where:

  - Protected P includes individuals that are removed from the susceptible pool and are thus protected from the virus. This can happen through full isolation as in China in early 2020 [35] or by different vaccination strategies which reduce the susceptible pool;

  - Quarantined Q describes latent carriers that are identified and quarantined after monitoring and tracing of contacts.

We do not explicitly introduce a second quarantined state for isolation of confirmed cases after the Infectious state [17, 36] but consider this together with the Removed state, by tuning the removal rate with an extra parameter (see [37] and references therein). Quarantining infected symptomatic patients is a necessary first step in every epidemic [38]. An additional link from Q to R, even though realistic, is neglected as both compartments are already outside the "contagion system" and would therefore be redundant from the perspective of evolution of the infection. In general, protected individuals can get back to the pool of susceptible after a while, but here we neglect this transition, to focus on simulating mitigation programs alone at their early stage. Long-term predictions could be modelled even more realistically by considering such link, that would lead to an additional parameter to be estimated and is beyond the scope of the present paper.

The model has in total 7 parameters. Three of them ($\beta$, $\alpha$, $\gamma$ introduced in Fig 1) are based on the classical SEIR model. The new parameters $\rho$, $\mu$, $\chi$, $\eta$ account for alternative mitigation programs to control the infectious curve (see Table 2 for details). Commonly, social distancing is modelled by the parameter $\rho$. In a closed-system setting where all individuals belong to the susceptible pool, but interact less intensively with each other, $\rho$ tunes the contact rate parameter $\beta$, resulting in the effective reproduction number $\hat{R} = \rho \cdot \beta\gamma^{-1}$. The parameter $\mu$ stably decreases the susceptible population by introducing an active protection rate. This accounts for improvements of public health, e.g. stricter lockdown of communities, or reduction of the pool of susceptible people after reduced commuters' activity, or vaccination. The parameter $\chi$ introduces an active removal rate of latent carriers. Intensive early contact tracing and improved methods to detect asymptomatic latent carriers may enhance the removal of exposed subjects from the infectious network. Following earlier works [39, 40] and adjusting the current parameters, $\hat{R}$ can be then expressed as $\hat{R} = \beta\gamma^{-1}\alpha(\alpha + \chi)^{-1}$. Finally, $\eta$ models the isolation of contagious individuals by handling the removal rate. This would correspond to identifying infectious individuals before they recover or die, and prevent them from infecting other susceptibles. Consequently, for this parameter alone $\hat{R} = \beta(\gamma + \eta)^{-1}$. Parameter values that are not related to mitigation strategies are set from COVID-19 epidemic literature [37, 41], as the main focus of the present model lies on sensitivity analysis of mitigation parameters. Our model can be further extended by time dependent parameters [38]. Default values for mitigation parameters are $\{\rho, \mu, \chi, \eta\}$ = $\{1,0,0,0\}$, corresponding to the classical SEIR model.

The dynamics of our SPQEIR model is described by the following system of differential equations:

$$\dot{S} = -\frac{\rho\beta SI}{N} - \mu S \ ,$$
$$\dot{E} = \frac{\rho\beta SI}{N} - (\chi + \alpha)E \ ,$$
$$\dot{I} = \alpha E \ - (\gamma + \eta)I \ ,$$
$$\dot{R} = (\gamma + \eta)I \ ,$$
$$\dot{P} = \mu S \ ,$$
$$\dot{Q} = \chi E \ ,$$

Here, $\dot{N} = 0$ with N = S + E + I + R + P + Q, implying the conservation of the total number of individuals. As value for the qualitative study, we used N = 10,000. For the cross-country assessment, N is adjusted to true population values for each country. Overall, the effective reproductive number becomes

$$\hat{R} = \frac{\rho\beta}{\gamma + \eta}\frac{\alpha}{\alpha + \chi}\frac{S}{N}, \tag{1}$$

Mitigation measures are initiated several days after the first infection case. Hence, we activate non-default parameter values after a delay $\tau$. For data fitting, we fit and compare $\tau$ to the official date when measures are initialized (cf. Table 1). To integrate the model numerically, we use the *odeint* function from *scipy.integrate* Python library.

## Model fitting

To show how our conceptual analysis is able to reproduce and explain empirical data, we fit the model to the official number of currently infected (active) cases of the first epidemic wave (winter-spring 2020), for each considered country. The choice is corroborated by the fact that all considered countries applied rapid, population-wide measures [24]. Model fitting to the

**Table 1. Test countries, with measures implemented.**

| Country | Measures | Param. involved | Starting Date | Population (rounded) |
|---|---|---|---|---|
| Austria (AT) | Partial lockdown | $\mu, \rho$ | 16 Mar | 9,000,000 |
| | Social distancing | $\rho, \mu$ | 16 Mar | |
| | Contact tracing | $\chi$ | 16 Mar | |
| | Phase-out | | Around 14 April | |
| Denmark (DK) | Social distancing | $\rho, \mu$ | 13 Mar | 6,000,000 |
| | Mild surveillance | $\eta$ | 13 Mar | |
| | Phase-out | | 14 Apr | |
| Ireland (IR) | Partial lockdown | $\mu, \rho$ | 28 Mar | 5,000,000 |
| | Social distancing | $\rho, \mu$ | 13 Mar | |
| | Phase-out | | 18 May | |
| Israel (IL) | Partial lockdown | $\mu, \rho$ | 15 Mar | 9,000,000 |
| | Social distancing | $\rho, \mu$ | 15 Mar | |
| | Contact tracing | $\chi$ | 15 Mar | |
| | Phase-out | | 19 April | |
| Lombardy (LO) | Lockdown | $\mu, \rho$ | 13 Mar (Italian) | 10,000,000 |
| | Social distancing | $\rho, \mu$ | 13 Mar | |
| | Phase-out | | Around 15 Apr | |
| Switzerland (CH) | Lockdown | $\mu, \rho$ | 16 Mar | 8,500,000 |
| | Social distancing | $\rho, \mu$ | 16 Mar | |
| | Phase-out | | 27 Apr | |

Test countries, with corresponding implemented measures (following the ACAPS database [24]), parameters in our SPQEIR model, starting date and rounded population of each country. For Lombardy, we used the Italian official date for lockdown. Ireland issued measures on two different dates; we use this case to compare social distancing and lockdown effect in a single country. We assume that the parameter $\eta$ is associated to all countries, which worked to isolate contagious individuals.

infectious curves is performed in two steps, using the parameters known to be active (cf. Table 1). First, we estimate the "model consistent" date of first infection, so that the simulated curve matches the reported data of active infections. This initial step corresponds to setting the time initial conditions of the SEIR model [17]. The fitting is performed with default parameter values, on a subset of data corresponding to the first outbreak, from first case until when measures are implemented (cf. Table 1). We use a grid search method for least squares, sufficient to fit a single parameter:

$$t_0 = \left\{ t' \mid RMS = \min_{t'} \sqrt{\frac{\sum_{i=t'}^{t_m} (x(i) - \hat{x}(i))^2}{n}} \right\} \tag{2}$$

where $t_0$ is the "model consistent" estimated date of first infection, $t_m$ refers to the date measures are implemented, $\hat{x}$ and $x$ are respectively reported and model-predicted data, and $n$ is the number of points between $t$ and $t_m$.

The second step estimates a reasonable set of the mitigation parameters that yield the best fitting of the simulated SPQEIR curve on reported data, during the first phase with implemented measures. This period is identified between the starting date $t_m$ (also included in the fitting) and the phase-out date $t_p$, cf. Table 1. Holding the epidemic parameters to literature values to achieve cross-country comparison on intervention parameters alone, the fitting is performed for a set of mitigation parameters relative to each country, as reported by policy

databases (cf. Table 1). The fit is performed with the widely used *lmfit* Python library. In S1 Text, we discuss such fitted parameters set and alternative ones.

We also perform a comparative quantitative analysis between our extended model and the simplest SEIR that lumps parameters under a single "social distancing" $\rho$. This allows comparing the estimate reproduction number $\hat{R}$ and shows the similarity or divergence of different control strategies in explaining the data. To assess how well they allow to fit the data, we employ the classical reduced $\chi^2$ statistics to evaluate the goodness-of-fit for each of the two models, considering the degrees of freedom [43]:

$$\chi^2_{red} = \frac{1}{n'-1-k} \sum_{j=1}^{n'} \frac{(y_j - \hat{y}_j)^2}{\hat{y}_j} \tag{3}$$

where $n'$ is the number of data points until phase-out, $k$ is the number of parameters in the model, $y_j$ are estimated values (from data) and $\hat{y}_j$ the expected ones (from model simulations).

## Results

First, we focus on the conceptual analysis of the effect of preventive mitigation interventions, initially for single measures (social distancing, active protection and active quarantining) and subsequently for a number of synergistic approaches. Additionally, we compare them to the effect of isolating contagious individuals. In particular, we study how crucial quantities, namely $\hat{R}$, the infectious peak height and time to zero infectious $\mathbb{T}$, depend on mitigation parameters. We define $\mathbb{T}$ as the time when there are less than 0.5 individuals in the I compartment, because ODE models approximate discrete quantities with continuous variables. Finally, we perform model fitting and intervention assessment over a set of countries. This provides quantitative outputs about the effectiveness of control measures, informing about the synergies applied and enabling cross-comparison.

### Simulations of single suppression measures

**Only social distancing.** The parameter $\rho$ captures social distancing effects, taking values in the interval [0, 1], where 0 indicates no contacts among individuals while 1 is equivalent to no action taken. To perform the current simulations, we assume a delay $\tau$ in implementing the measures of 10 days. Such value does not modify the qualitative behavior of the epidemic dynamics but influences the quantitative estimations of peak height and mitigation timing. We refer to S1 Text for further discussion. Overall, Fig 2 reports simulation results about the

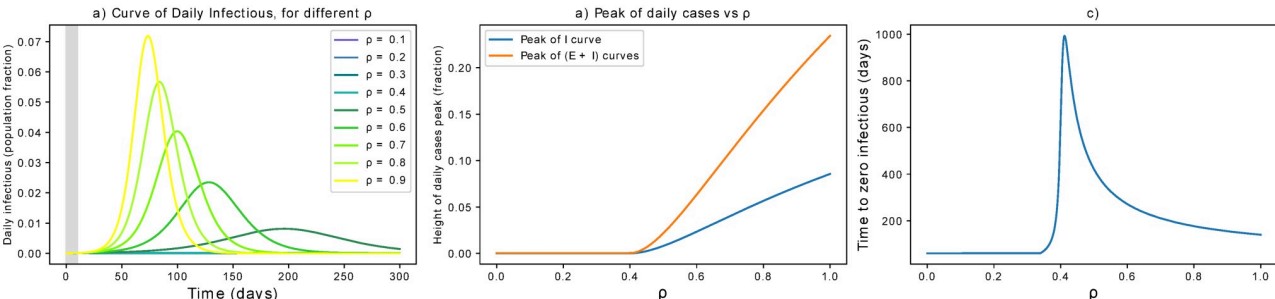

**Fig 2. Effect of social distancing.** (a) Effects of social distancing on the epidemic curve. The grey area indicates when measures are not yet in place. (b) The peak is progressively flattened until a mitigation is reached for sufficiently small $\rho$. For these settings, the critical value for $\rho$ is 0.4 (it pushes $\hat{R}$ below 1). (c) Unless $\rho$ is small enough, stronger measures of this kind might delay the mitigation time $\mathbb{T}$ of the epidemic.

effects of $\rho$. The curve of infectious is progressively flattened by social distancing (Fig 2a) and its peak mitigated (Fig 2b). However, the time to mitigation gets delayed for decreasing $\rho$, until a threshold yielding a disease-free equilibrium rapidly (Fig 2c). In this case, the critical value for $\rho$ is 0.4, leading to $\hat{R} < 1$. Fig 2c reveals that the dependence of $\mathbb{T}$ on $\rho$ is not monotonous. With the current settings, values of $\rho \leq 0.3$ are best effective to minimise the mitigation timing. In general, the optimal $\rho$ value that minimises mitigation timing depends on $\tau$, as discussed in S1 Text. In fact, longer delays in issuing interventions are not only associated with higher peaks in the infection curves, but also in more stringent parameter values that are necessary to obtain minimal $\mathbb{T}$. This fact further stresses the importance of prompt interventions to control the quantitative aspects of epidemic mitigation.

**Only active protection.** As discussed above and in S1 Text, our simulations take into account 10 days delay from the first infection to the initiation of active protection. Small values can reflect continuous improvement of protection measures (as people learnt better how to deal with the virus) or different vaccination strategies (thus going beyond non-pharmaceutical strategies). Higher values are considered to model certain effects of a step-wise hard lockdown (see following paragraph). The results are reported in Fig 3. We see that small precautions can make an initial difference (Fig 3a and 3b). The time to zero infectious is decreased with higher values of active protection (Fig 3a and 3b). In particular, $\mu = 0.01 d^{-1}$ mitigates the epidemics in about 6 months by protecting 70% of the population. Higher values of $\mu$ achieve mitigation faster, while protecting almost 100% of the population. It is probably not fully realistic to consider that these protection rates are obtained only by isolation. Instead, they could represent improved hygiene routines or vaccination strategies and are thus worthy to consider.

In addition to what analysed above, we also consider strategies which isolate many people at once [44]. This corresponds to reducing S to a relatively small fraction rapidly. Since $\mu$ is a rate, we mimic what could happen during a step-wise hard lockdown: large values of $\mu$, but whose effect only lasts for a short period of time (Fig 4b). We thus use the notation $\mu_{ld}$. In the figure, an example shows how to rapidly protect about 68% of the population with a step-wise $\mu_{ld}$ function. In particular, we use an average four-days long step-wise $\mu_{ld}$ function (Fig 4b) to mimic the rapid, but not abrupt, change in mobility observed in many countries by Google Mobility Reports [30] (Fig 4c). The effects of strong, rapid protection are reported in Fig 4a, showing that such strategy is effective in mitigating the epidemic curve and in reducing the time to mitigation.

**Only active quarantining.** Controlling latent carriers before symptom onset is an important strategy to limit transmission. We here consider how mitigation is achieved by targeted interventions, e.g. by contact tracing, and we quantify the interplay between precision and

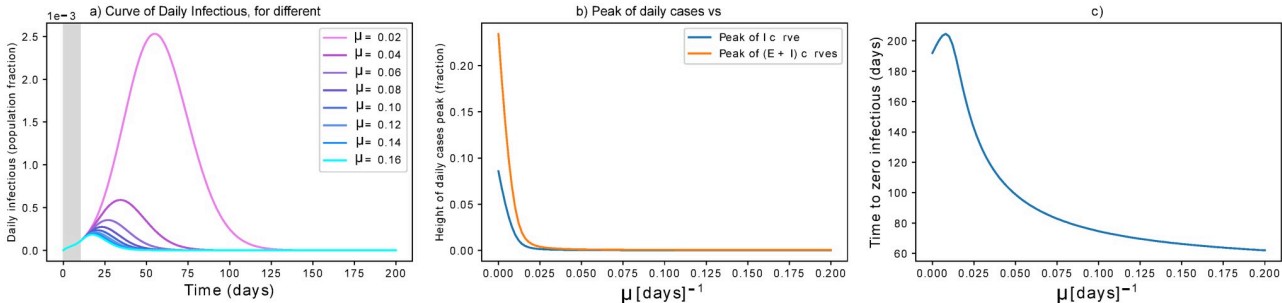

**Fig 3. Effect of lock-down.** (a) Effects of active protection on the infectious curve. The grey area indicates when measures are not yet in place. $\mu$ is expressed in $d^{-1}$. (b) Dependency of peak height on $\mu$: the peak is rapidly flattened for increasing $\mu$, then it is smoothly reduced for higher parameter values. (c) High $\mu$ values are effective in anticipating the mitigation of the epidemic, but require protecting more than 90% of the population.

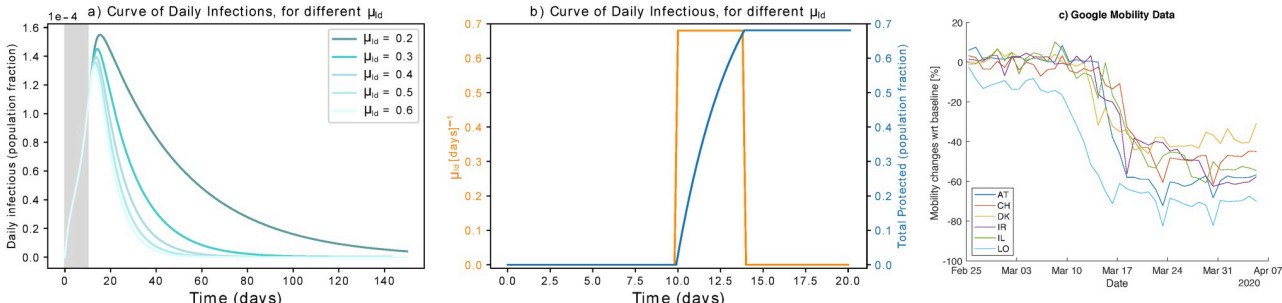

**Fig 4. Effect of step-wise hard lock-down.** (a) Flattening the infectious curve by hard lockdown. Rapidly isolating a large population fraction is effective in mitigating the epidemic spreading. (b) Modeling hard lockdown: high $\mu_{ld}$ (orange) is active for four days to isolate and protect a large population fraction rapidly (blue). As an example, we show $\mu_{ld} = 0.28 d^{-1}$ if $t \in [10, 14]$. It results in protecting about 68% of the population in two days. Higher values, e.g. $\mu_{ld} = 0.65 d^{-1}$ would protect 93% of the population at once. (c) Google Mobility Report visualization [30] for analysed countries, around the date of measures setting. Each line reports the mean in mobility change across Retail & Recreation, Grocery & Pharmacy, Transit stations, and Workplaces, around the date of implementation of the measures. A minimum of four days (from top to bottom of steep decrease) is required for measures to be fully effective. Abbreviations explanation: AT = Austria, CH = Switzerland, DK = Denmark, IL = Israel, IR = Ireland, LO = Lombardy.

delay in tracing, thus expanding [45]. As above, not only we consider the impact on $\hat{R}$ but on the whole infectious curve, its height and its time evolution.

The simulations in this part are based on realistic assumptions: testing a person is effective only after a few days that that person has been exposed (to have a viral charge that is detectable). This induces a maximal quarantining rate $\theta$, which we set $\theta = 0.33 d^{-1}$ as testing is often considered effective after about three days from contagion [46]. Therefore, we get the active quarantining rate $\chi = \chi' \cdot \theta$, where $\chi'$ is a tuning parameter associated e.g. to contact tracing. As $\theta$ is fixed, we focus our analysis on $\chi'$. As above, we also assume that testing starts after the epidemic is seen in the population, i.e. some infectious are identified with 10 days delay in the activation of measures.

The corresponding results are reported in Fig 5. The curve is progressively flattened by latent carriers quarantining and its peak mitigated, but the time to mitigation gets delayed for increasing $\chi'$. This happens until a threshold value of $\chi'_{thr} = 0.9$ that pushes $\hat{R}$ below 1. This value holds if we accept a strategy based on testing, with $\theta = 0.33$. If preventive quarantine of suspected cases does not need testing (for instance, when it is achieved by contact tracing apps), the critical $\chi'$ value could be drastically lower. In particular, $\chi'_{thr} = 0.3 \, d^{-1}$ if $\theta = 1 d^{-1}$, i.e. latent carriers are quarantined the day after a contact.

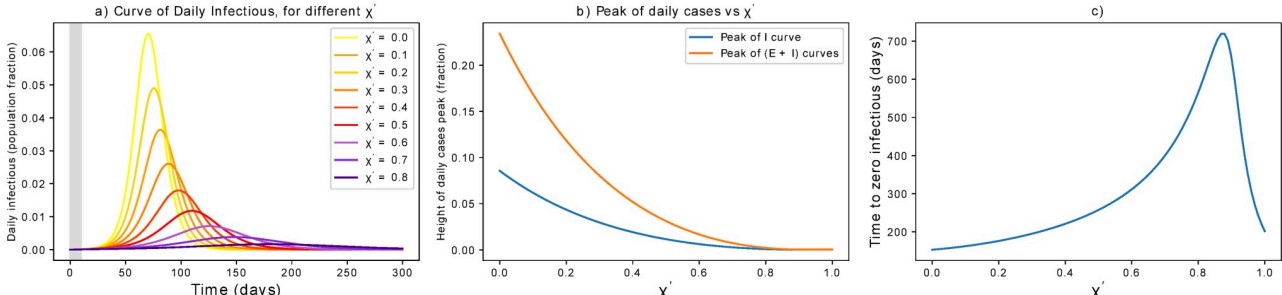

**Fig 5. Effect of latent carriers quarantining.** (a) Effects of active latent carriers quarantining on the epidemic curve. The grey area indicates when measures are not yet in place. (b) The peak is progressively flattened until a disease-free equilibrium is reached for sufficiently large $\chi'$. (c) Unless $\chi'$ is large enough, stronger measures of this kind might delay the mitigation of the epidemic. Note that the critical $\chi'$ can be lowered for higher $\theta$, e.g. if preventive quarantine does not wait for a positive test.

The parameter $\chi'$ tunes the rate of removing latent carriers. Hence, it combines tracing and testing capacities, i.e. probability of finding latent carriers ($P_{find}$) and probability that their tests are positive ($P_+$). The latter depends on the false negative rate $\delta_-$ as

$$P_+ = (1 - \delta_-). \tag{4}$$

So, $\chi' = P_{find} \cdot P_+$. Hence, mitigating the peak of infectious requires an adequate balance of accurate tests and good tracing success as reported in Fig 6. Further quantifying the latter would drastically improve our understanding of the current capabilities and of bottlenecks, towards a more comprehensive feasibility analysis.

**Only isolation of infectious.** Isolating contagious individuals is a first step to contrast the pandemic, on top of preventive measures. In this section, we consider its effect alone, to be compared with that of other single parameters shown above. As discussed above, we here consider simulations that include a delay of 10 days from the first infection to the initiation of the measures. Quantitative changes associated with different $\tau$ are discussed in the S1 Text. The results are reported in Fig 7. Targeting the infectious population means that fewer people can spread the contagion. The curve of infections is progressively flattened, the more rapidly contagious people are identified and isolated, until a threshold value $\eta = 0.51$ (for our initial parameters). In turn, the mitigation time gets longer if $\eta$ is increased, but has not yet crossed the threshold value. These findings point to the importance of complementing the control of contagious individuals with additional preventive measures such as the ones presented above. We acknowledge that these results are valid on average, but that breaking the infectious chain at specific links can have additional benefits in heterogeneous social networks.

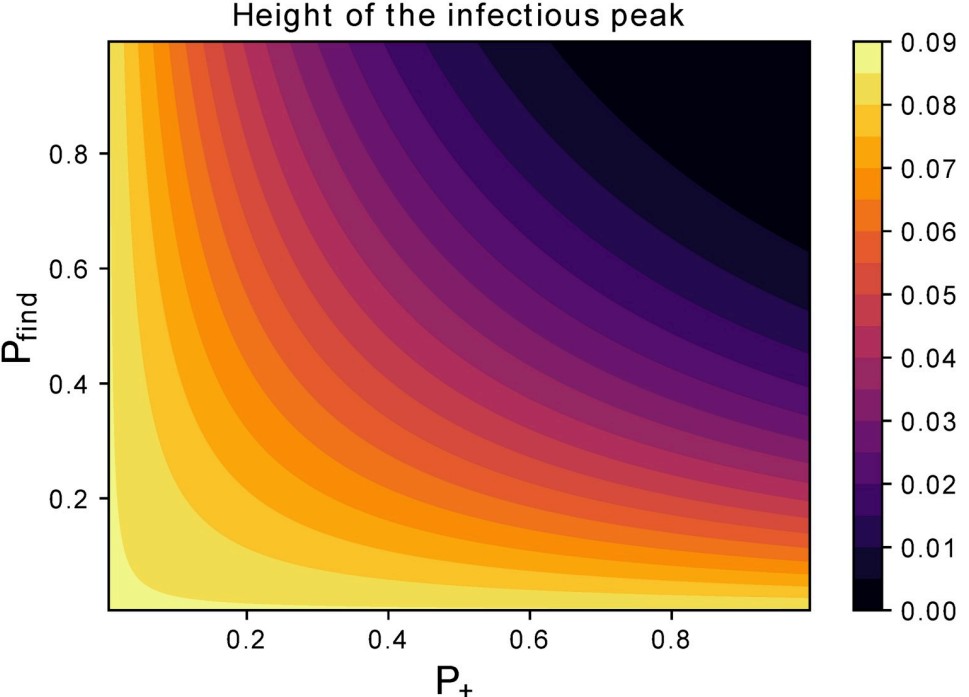

**Fig 6. Dependence of latent carriers quarantining on control parameters.** Assessing the impact of $P_{find}$ and $P_+$ on the peak of infectious separately. This way, we separate the contribution of those factors to look at resources needed from different fields, e.g. network engineering or wet lab biology. Solutions to boost the testing capacity like [47] could impact both terms.

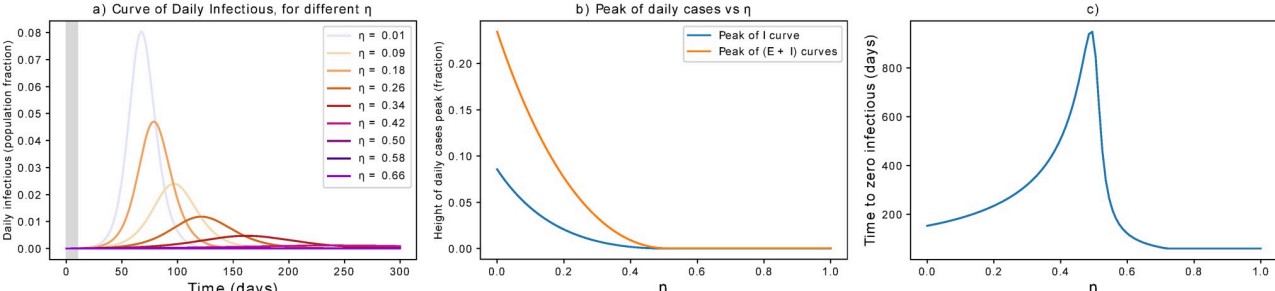

**Fig 7. Dependence of infectious isolation on control parameters.** (a) Effects of isolation of contagious individuals on the epidemic curve. The grey area indicates when measures are not yet in place. (b) The peak is progressively flattened until a disease-free equilibrium is reached for sufficiently large $\eta$. (c) Unless $\eta$ is large enough, stronger measures of this kind might delay the mitigation of the epidemic. Note that the critical $\eta$ can be higher if there is delay in intervening, i.e. if infectious individuals are isolated after several days and can thus spread the infection.

## Synergistic scenarios

Fully enhanced active quarantining and active protection might not be always feasible, e.g. because of limited resources, technological limitations or welfare restrictions. On the other hand, the isolation of a limited portion of contagious individuals could not be sufficient. Therefore a synergistic approach is very attractive as it can flatten the curve with combinations of interventions that target different population groups and require distinct resources. This section shows a number of possible synergies, concentrating as before on abstract scenarios to investigate how combining different mitigation programs impact the control parameter $\hat{R}$ (*cf.* Eq (1)), the infection curves and the mitigation timing.

As case studies, we consider the 6 synergistic scenarios listed below. Parameters are set without being specific to real measures taken: their value is so far conceptual and meaningful when compared across scenarios. Just like above, we consider a 10 days delay from the first infection to issuing measures; as suggested in other studies [48], delaying action could worsen the situation. To differentiate between a rapid isolation and a constant protection, we use $\mu_{ld}$ (associated to "hard lockdown strategies", see Section "Only active protection") separated from $\mu$. To get $\hat{R}$ when measures are initiated, we follow Eq 1, considering $\chi = \chi' \cdot \theta$ as in Section "Only active quarantining". Our scenarios are the following:

1. During the first COVID-19 wave, many European countries opted for a lockdown strategy. A quite large fraction of the population was isolated, individuals were recommended to self-quarantine in case of suspected positiveness, social distancing got mandatory but was sometimes not fully followed, masks and sprays were suggested for protection. So, we set an initial "rapid protection" $\mu_{ld} = 0.12$ to protect around 38% of the population quickly. Then we chose $\rho = 0.7$, $\chi' = 0.12$, $\eta = 0.12$ and $\mu = 0$. This yields $\hat{R} = 0.65$.

2. In case that isolation of contagious individuals fails, an alternative procedure is to rapidly protect only the population fraction at high risk ($\mu_{ld} = 0.06$, driving 15% of initial S to P). Social distancing and latent carrier quarantine should then be enforced ($\rho = 0.65$, $\chi' = 0.55$). This gives $\hat{R} = 0.67$.

3. In case both preventive quarantine of latent carriers and isolation of contagious are not greatly effective ($\chi' = 0.03$, $\eta = 0.07$), and in case of low protection rate and scarce isolation ($\mu = 0$, $\mu_{ld} = 0.08$), we rise social distancing for all individuals doing business as usual ($\rho = 0.45$). In this case, $\hat{R} = 0.64$.

4. If there are no safety devices that provide an adequate protection ($\mu = 0$) and no isolation is foreseen ($\mu_{ld} = 0$), we set $\rho = 0.6$, $\chi' = 0.2$, $\eta = 0.25$ to get $\hat{R} = 0.65$.

5. This case has higher $\hat{R}$ than the previous ones, namely $\hat{R} = 0.84$. The corresponding parameters are $\mu_{ld} = 0.1$, $\mu = 0.002$, $\rho = 0.7$, $\eta = 0.1$. This shows that even low enforcement of single interventions can achieve $\hat{R} < 1$, even thought the corresponding mitigation is slower.

6. Finally, we consider "draconian" [49] measures such that $\hat{R} = 0.32$ only through isolation and massive screening, that targets Exposed and Infectious individuals. So, $\mu_{ld} = 0.3$, $\chi' = 0.1$, $\eta = 0.2$ while $\rho = 1$ and $\mu = 0$. This points to the importance of tracing capacities to min-imise the total isolation period.

Simulation results in Fig 8 show that different synergies can lead to different timing, even though the peak is contained similarly (Fig 8a). This has an impact on the cumulative number of cases (Fig 8b) that will be reflected on the death toll. This holds even when the $\hat{R}$ values are very close, as in scenarios 1 to 4: even though $\hat{R}$ is the main driver of the epidemic, the contri-bution of finer-grained parameters is relevant for the fine-tuning of interventions. Focusing on scenarios 2 and 3, we notice that prevention measures and latent quarantine accelerate the mitigation, even when isolating only vulnerable people. This achieves similar effects as strong social distancing. In addition, active protective measures with relatively low values further con-cur in mitigating the peak. This finding asks for rapid assessment of masks and sanitising routines.

Overall, the strength of mitigation measures influences how and how fast the epidemic is flattened. $\mu_{ld}$ mostly governs the peak height after measures are implemented, $\rho$ mainly tunes the curve steepness together with $\mu$, while $\chi$ shifts the decaying slope up and down. Overall, a $\hat{R} < 1$ suffices to avoid breakdown of the health system, but its effects could be too slow. Decreasing its value with additional synergistic interventions could speed up epidemic mitiga-tion. A careful assessment of measures' strength is thus recommended for cross-country comparison.

## Model fitting and interventions assessment

In this section, we test our results on several datasets, to estimate the likely impact of different strategies and to show which combination could have yield a similar $\hat{R}$. This way, we show

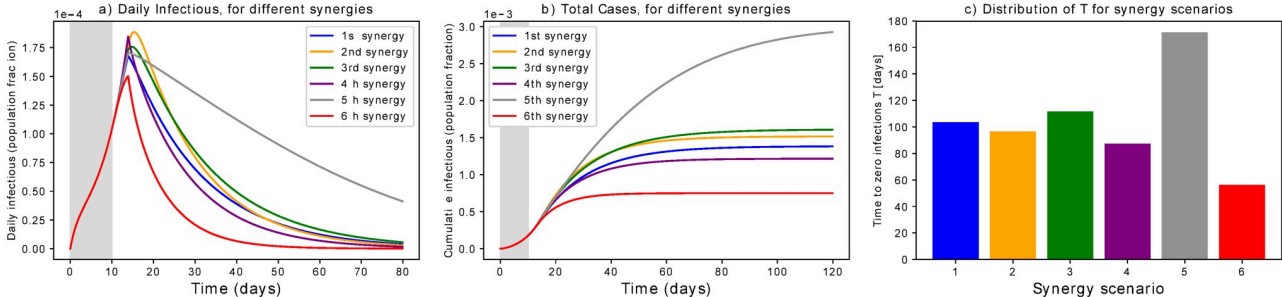

**Fig 8. Synergistic scenarios.** Simulations of the 6 synergistic scenarios. (a) Curves of infectious Individuals, (b) Cumulative cases. The grey area indicates when measures are not yet in place. It is evident that scenarios leading to similar $\hat{R}$ could show different patterns and mitigation timing. (c) Distribution of times to zero infections $\mathbb{T}$ for different scenarios.

**Table 2. SPQEIR model parameters.**

| Fixed parameters | Mitigation parameters | |
|---|---|---|
| $\beta$ = (average contact rate in the population) = 0.85 $d^{-1}$ | $\mu$ = (rate of active protection) $[d^{-1}]$ | |
| $\alpha$ = (mean incubation period)$^{-1}$ = 0.2 $d^{-1}$ | $\rho$ = (social distancing tuning) | |
| $\gamma$ = (mean infectious period)$^{-1}$ = 0.34 $d^{-1}$ | $\chi$ = (active removal rate) $[d^{-1}]$ | |
| $R_0$ = 2.5 | $\eta$ = (rate of contagious isolation)$[d^{-1}]$ | |

SPQEIR model parameters with their standard values for the COVID-19 pandemic from literature [37, 42]. Here "*d*" denotes days.

how countries could achieve mitigation through a synergy of control measures with similar impact on the epidemic but different management and possibly socio-economic impact.

**Model fitting.** As described in the Methods section, we first estimate the "model consistent" date of first infection $t_0$, *i.e.* the temporal initial condition for the SPQEIR model. Comparing this date with the starting date for intervention measures (Table 1) corresponds to estimating $\tau$ for each country. We do not claim this to be the true date of first infection in a country; it is the starting date of infections in case of homogeneous transmission, under the assumption of no superspreading events [50], and with the hypothesis of coherent $R_0$ (cf. Table 2). During the second fitting step, we also estimate the date at which mitigation measures start having effect on the infectious curve, $t_m$. Comparing $t_m$ with official intervention dates from Table 1, we notice that about 8 days are necessary to register lockdown effects. This is consistent to early findings on lockdown effectiveness [51]. Estimated dates are reported in Table 3.

Then, we fit mitigation parameters to data of active cases, from the estimated starting date of control measures $t_m$ to phase-out $t_p$ (*cf.* Table 1). The active parameters for the fit are reported in the same table. For the protection parameter, we used $\mu_{ld}$ acting on 4 days (as introduced in Fig 4) since it better reflects the rapid protection of certain individuals that happened during the first COVID-19 wave. Since $S \sim N$, its quantitative impact is anyway greater on the $S$ compartment. Other compartments are impacted by the remaining parameters. The results of the model fitting are reported in Fig 9. The SPQEIR model, with appropriate parameters for each country (cf. Table 1), is fitted to reported infection curves and, overall, model fitting have good agreement with data. This supports the model structure as very simple yet realistic enough to capture the main dynamical behaviour of the infection curves in multiple countries. In addition, it allows for each country to obtain multiple sets of parameters representing different strategies. We notice that the effect of social distancing ($\rho$) is predominant as it homogeneously prevents the big pool of Susceptible individuals to stream into the Exposed compartment. However, also tracing and isolation can have a considerable effect in complementing population-wide interventions. The values associated to the fitted parameters

**Table 3. COVID-19 significant dates.**

| Country | AT | DK | IR | IL | LO | CH |
|---|---|---|---|---|---|---|
| 1$^{st}$ official detection | 24 Feb | 04 Mar | 29 Feb | 21 Feb | 21 Feb | 25 Feb |
| $t_0$ | 22 Jan | 22 Jan | 29 Jan | 24 Jan | 05 Jan | 14 Jan |
| $t_m$ | 26 Mar | 21 Mar | 06 Apr | 30 Mar | 19 Mar | 21 Mar |

Dates of official detection of first COVID-19 case [25], estimated dates for first infection $t_0$ (according to Eq 2) and date at which measures start being effective $t_m$, per country.

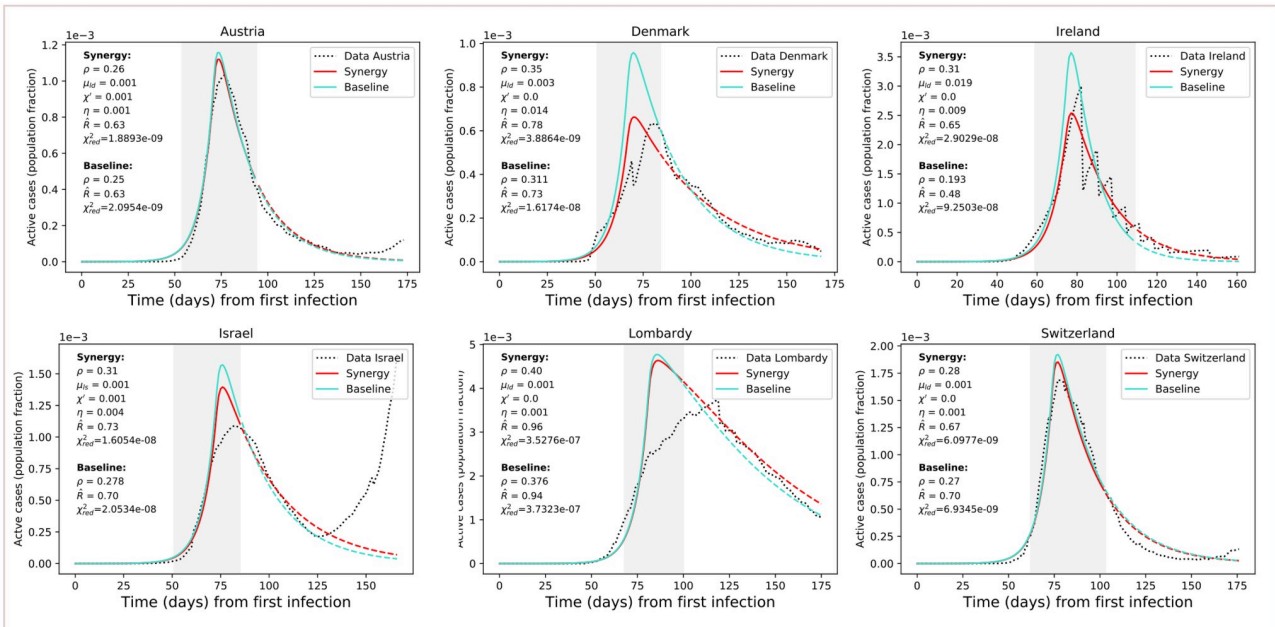

**Fig 9. Results of model fitting.** Results of model fitting. Infection curves for the considered countries (dotted) are fitted with the SPQEIR model with appropriate parameters (red curves). We also show a comparison with the fitted curve obtained from the "basic" SEIR model with only social distancing (turquoise curves). Parameter values are reported for each country, as well as the corresponding $\hat{R}$ (for the grey area, following Eq 1) and $\chi^2_{red}$. The period of measures enforcement, from $t_m$ to $t_p$, is highlighted by the grey region. Time progresses from the estimated day of first infection $t_0$ (cf. Table 3). Population fraction refers to country-specific populations (cf. Table 1). After phase-out, we prolong the fitted curve (parameter values unchanged) to compare observed data with what could have been if measures had not been lifted (dashed lines). From the data, we can observe a resurgence of cases that points to possible "second outbreaks" (particularly in Israel).

correspond to non-negligible numbers of individuals affected by the interventions. Such values are discussed in S1 Text. This is informative about how synergistic approaches can realistically explain the mitigation of the infectious curve, and highlight the potential advantages associated with modifying the combination of strategies in subsequent epidemic waves. Finally, it allows a comparison between different countries through the corresponding best fit parameters. For Ireland, although initial social distancing advises were issued on 13th March (cf. Table 1), fitting the complete curve was only possible when considering the lockdown date (28th March) as the major driver of the mitigation.

Model fitting is slightly hindered by data quality. For instance, Ireland reported intermittent data, while Lombardy is not perfectly represented, probably because of some data reporting issues and larger heterogeneity in its spacial patterns.

Finally, the reduced $\chi^2$ metric (Eq 3) reports that the complete SPQEIR model and the simple social distancing one attain similar goodness of fit, although values for the SPQEIR are slightly lower in all cases. Country-specific extra parameters (*cf.* Table 1) are thus useful to fine-tune the reproduction of epidemic curves, as noticed in the conceptual analysis, Sec. "Synergistic scenarios". This shows that synergistic measures are able to provide a similar mitigation of the curve of infections, and an analogous $\hat{R}$, as the pure social distancing scenario. In turn, synergistic approaches allow lower social distancing values, possibly having less severe social and psychological impacts on the population. This in turn supports the use of several interventions to control the epidemic curve in an effective and timely manner, while balancing social benefits. In addition, the SPQEIR is confirmed to be informative, on top of being fully interpretable and linked to recognised social policy categories.

**Cross-country interventions assessment.** Fitting a number of countries with the same model containing the same epidemiological parameters allows to perform a comparison on the efficacy of their interventions, to inform future decision making. In Fig 9, parameter values providing the best fit of model to data are reported, together with the simulation results (mean values) calculated by the *lmfit* algorithm [52]. Different synergies yield similar values for $\hat{R}$, but the curve is different in its evolution as already observed in the previous sections. As expected from the model analysis above, the lower $\hat{R}$ is (below 1), the faster the mitigation of the epidemic. In addition, different parameter combinations generate curves that differ in amplitude and time evolution. This might well explain differences in reported total cases and deaths between various countries. Comparing Austria, Denmark and Lombardy, we observe that contact tracing and monitoring contribute to speeding up the curve decay, despite the fact that population-wide interventions played so far a major role. In general, combined isolation and tracing strategies would reduce transmission in addition to social distancing or self-isolation alone. In general, a strong, rapid lockdown that combines protection and social distancing seems the best option, as also suggested by the conceptual analysis. However, intervening with additional synergies is a viable option to mitigate the epidemic faster and with lower social values.

Finally, we observe the value of timely interventions: we see that intervening earlier with respect to the date of first infection helps reducing the daily curves by almost a factor of 10. For instance, we can compare Denmark and Lombardy in Fig 9: the first one got a peak corresponding to about 0.08% of the whole population, while the second region registered a number of active cases of about 0.5% of the whole population. This translates in more than 3800 infectious on the Danish peak, and on more than 37000 on Lombardy's.

## Discussion

The SPQEIR model assesses and compares the effectiveness of several control measures to mitigate the COVID-19 epidemic curve. It integrates previous literature and considers synergy strategies often considered alone. In particular, we focus on preventive measures, i.e. those that target people that are not yet fully infectious. Initially, we perform conceptual simulations to investigate the effect of single and combined measures not only on $\hat{R}$, but also on the complete time evolution of the infectious curve. Then, we compare them with the isolation of contagious individuals. The possibility of choosing among several strategies is of practical importance for decision makers: a comparison of Figs 2, 3, 5 and 7 reveals that increasing social distancing delays and decrease the height of the peak of infections, increasing active protection as well decreases the height of the peak of infections, but anticipates the occurrence of such peak, increasing active quarantining also delays and decrease the height of the peak of infections like social distancing, but the same peak mitigation by active quarantining is associated with shorter delays than with social distancing.

Moreover, the model is fitted to several countries, to estimate the plausible impact of synergistic strategies. The fit is performed until phase-out dates for the first epidemic wave (winter-spring 2020), when measures are progressively lifted and therefore the model assumptions do not hold anymore. We remark that the current set of parameters may not be unique, as there is high correlation among parameters. This is a common identifiability issue of SIR parameters, particularly when several of them contribute to the same control parameter $\hat{R}$ [53]. For instance, the *lmfit* diagnostic reports 0.9 correlation between $\rho$ and $\mu_{ld}$ and 0.99 between $\rho$ and $\chi$, for Austria. This means that they can equally well explain the evolution of the curve, so they could be alternatively chosen for epidemic control, while targeting different population groups. This is in line with our above analysis, as we aim at showing how different

combinations of interventions can tune the mitigation of infection curves. We remark that, due to this degeneracy of parameters (*i.e.* several combination can yield to same $\hat{R}$), the ones reported in Fig 9 constitute a reasonable set obtained by an automatic least-square algorithm, but estimating their true values (inverse problem) needs to be complemented with alternative, targeted approaches.

In Fig 9 we extrapolate the model, with same parameter values, after phase-out (dashed lines), to compare observed data to the most optimistic scenario, where measures would not have been lifted. We observe that, up to July 8th, the infection curves mostly maintained an inertial decreasing trend: despite some fluctuations that make them generally higher than the best scenario, they kept on following a downward trend similar to that of the model. We speculate that this phenomenon is linked to changed behaviors, face masks [54] and improved sanitising practices that maintained social distancing values, as well as contact tracing practices issued by many countries along with the phase-out. However, some countries (Israel in particular, but also Austria) already showed a worrisome upward trend, eventually associated to a second outbreak [27]. As this is not a low probability event, we stress the usefulness of our analysis to prepare for future developments in pandemic progression.

It has been asked whether the peak of infections was reached because of herd immunity or because of interventions [55]. An added value of this study is to confirm that the peak of infection, for the considered countries, was not reached because of herd immunity. On the contrary, it is the effect of a number of mitigation measures that reduced the number of cases artificially. This should warn about the high numbers of people that are still susceptible.

We acknowledge the limitations of our analysis. Due to its structure and the use of ordinary differential equations, the model only accounts for average trends. However, it cannot reproduce fluctuations in the data, being them intrinsic in the epidemic, or from testing and reporting protocols that might differ among countries. The model focuses on initialization of measures that last for short-medium periods, as it does not include out-fluxes from the "safeguarded" compartments P and Q. This assumption is not completely realistic and we are aware that household infections concurred to a significant number of contagions. Like other studies [56], our simulations thus underestimate the disease burden coming from this source. However, the synergy with other parameters can retain the modulation of the dynamics posed by different behaviors. Overall, our model is used to assess the validity of control measures rather than to predict the complete evolution of the epidemic. Similarly, in order to concentrate on the generic control of infectious curves, we did not include further compartments about hospitalization, as they are already upstream with respect to the I compartment, nor we considered asymptomatic patients, that would not impact the main findings about synergistic mitigation. In addition, the constant nature of parameters used in this analysis allows good agreement between model and data when countries implemented rapid and strong measures point-wise in time, with little follow-ups. Further studies, with time varying parameters, could obtain more precise values. In the same way, transferring models from country to country requires fulfilling the same assumptions on model structure and basic hypothesis. This is shown by the different fitting performances, that suggest that a transfer is not always possible. The same fitting performance is often impacted by the data quality, related to monitoring, testing and reporting; despite our carefulness in selecting countries that had similar positive rates, there could be additional uncertainties to the parameter values that we estimated. Finally, we remark that the retrospective dates in Table 3 should be interpreted under the model assumptions: they could suggest that the first infection happened several weeks before the official detection, but they could as well be associated to the inherent identifiability limits of SIR-like parameters [57, 58].

In general, this study is not intended to make a ranking of country responses, nor to suggest that different strategies could have led to better outcomes. Contrariwise, it should be used as a methodological step towards quantitatively inquiring the effect of different intervention categories and of their combinations. It examines possible abstract scenarios and compares quantitative, model-based outputs, but it is not intended to fully represent specific countries nor to reproduce the epidemic complexity within societies. In fact, the model does not provide fine-grained quantification of specific interventions, e.g. how effective masks are in protecting people, how much proximity tracing apps increase $P_{find}$, how changes in behavior are associated with epidemic decline [59] and so on. We acknowledge that the new compartments cannot perfectly match policy measures, but are a reasonable approximation. Some real measures might also affect multiple parameters at once, e.g. safety devices and lockdown could impact both $\mu$ and $\rho$. Comparing results of this macro-scale model with those of complex, micro-scale ones [3] could inform researchers and policy makers about the epidemic dynamics and effective synergies to hamper it. Any conclusion should be carefully interpreted by experts, and the feasibility of tested scenarios should be discussed before reaching consensus.

## Conclusion

We have developed a minimal model to link intervention categories against epidemic spread to epidemiological model compartments. This allows quantitative assessment of non-pharmaceutical mitigation strategies on top of social distancing, for a number of countries. Strategies have different effects on epidemic evolution in terms of curve flattening and timing to mitigation. As with previous studies [31, 60], we have observed the need to enforce containment measures (i.e., detect and isolate cases, identify and quarantine contacts and at risk neighborhoods) along with mitigation (i.e., slow down viral spread in the community with social distancing).

By extending the classic SEIR model into the SPQEIR model, we distinguished the impact of different control programs in flattening the peak and anticipating the mitigation of the epidemic. Depending on their strength and synergy, non-pharmaceutical interventions can hamper the disease from spreading in a population. First, we performed a complete sensitivity analysis of their effects, both alone and in synergy scenarios. Then, we moved from idealised representations to fitting realistic contexts, allowing preliminary mapping of intervention categories to abstract programs. We verified that the model is informative in interpolating the infection curves for a number of countries, and performed cross-country comparison. We could then obtain model-based outputs on the strength of interventions, for a number of countries that respected the model assumptions. This provides better, quantitative insights on the effect of mitigation measures and their timing, and allows improved comparison.

Overall, this work could contribute to quantitative assessments of epidemic mitigation strategies. To tackle current epidemic waves, and against possible resurgence of contagion [61] (also cf. Fig 9), better understanding the effect of different non-pharmaceutical interventions could help planning mid- and long-term measures and to prepare preventive plans while allowing a relaxation of social distancing measures. In fact, this synergistic approach still remains of high importance in this second lockdown times, where countries still need to balance different non-pharmaceutical interventions to keep the infection at bay while complementing vaccination strategies and containing the impacts on other aspects of society.

## Supporting information

**S1 File.**
(TXT)

**S1 Text.**
(PDF)

# Acknowledgments

The authors thank the Research Luxembourg—COVID-19 Taskforce for mutual collaborations. They also thank two anonymous reviewers for helpful comments and suggestions that led to an improvement of this paper.

# Author Contributions

**Conceptualization:** Daniele Proverbio, Andreas Husch.

**Data curation:** Daniele Proverbio.

**Formal analysis:** Daniele Proverbio, Françoise Kemp, Stefano Magni, Andreas Husch, Atte Aalto, Laurent Mombaerts, Alexander Skupin, Christophe Ley.

**Investigation:** Daniele Proverbio, Françoise Kemp, Stefano Magni, Alexander Skupin, Jorge Gonçalves, Christophe Ley.

**Methodology:** Daniele Proverbio.

**Project administration:** Alexander Skupin, Jorge Gonçalves.

**Software:** Daniele Proverbio, Françoise Kemp, Jose Ameijeiras-Alonso.

**Supervision:** Stefano Magni, Andreas Husch, Alexander Skupin, Jorge Gonçalves, Christophe Ley.

**Validation:** Daniele Proverbio, Christophe Ley.

**Visualization:** Daniele Proverbio.

**Writing – original draft:** Daniele Proverbio, Françoise Kemp, Stefano Magni, Christophe Ley.

**Writing – review & editing:** Daniele Proverbio, Françoise Kemp, Stefano Magni, Andreas Husch, Atte Aalto, Laurent Mombaerts, Alexander Skupin, Jorge Gonçalves, Jose Ameijeiras-Alonso, Christophe Ley.

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
