## [Decision Letter · Decision Letter 0]

16 Dec 2020

PONE-D-20-26921

Dynamical SPQEIR model assesses the effectiveness of non-pharmaceutical interventions against COVID-19 epidemic outbreaks

PLOS ONE

Dear Dr. Proverbio,

Thank you for submitting your manuscript to PLOS ONE. After careful consideration, we feel that it has merit but does not fully meet PLOS ONE’s publication criteria as it currently stands. Therefore, we invite you to submit a revised version of the manuscript that addresses the points raised during the review process.

In particular, Referee #2 has raised a number of critical remarks regarding the model structure and the statistical analysis that require careful consideration.

We look forward to receiving your revised manuscript.

Kind regards,

Michele Tizzoni

Academic Editor

PLOS ONE

Journal Requirements:

2. Please note that in order to use the direct billing option the corresponding author must be affiliated with the chosen institute. Please either amend your manuscript to change the affiliation or corresponding author, or email us at plosone@plos.org with a request to remove this option.

Reviewers' comments:

Reviewer's Responses to Questions

**Comments to the Author**

1. Is the manuscript technically sound, and do the data support the conclusions?

Reviewer #1: Partly

Reviewer #2: Partly

2. Has the statistical analysis been performed appropriately and rigorously? 

Reviewer #1: I Don't Know

Reviewer #2: Yes

3. Have the authors made all data underlying the findings in their manuscript fully available?

Reviewer #1: Yes

Reviewer #2: Yes

4. Is the manuscript presented in an intelligible fashion and written in standard English?

Reviewer #1: Yes

Reviewer #2: Yes

5. Review Comments to the Author

Reviewer #1: The paper introduces in the SEIR model 3 different types of non-pharmaceutical interventions i.e. social distancing, quarantine of exposed individuals and generalized lock-down. After the outbreak of COVID pandemic several papers have been published in this direction e.g.

Das et al. medRXIV https://doi.org/10.1101/2020.06.04.20122580;

Tomas de-Camino-Beck. medRXIV https://doi.org/10.1101/2020.05.19.20106492;

Perkins et al. Bulletin of Mathematical Biology 82, 118 (2020);

Lai et al. Nature volume 585, 410–413 (2020);

Alrrashed et al Informatics in Medicine Unlocked 20, 100420 (2020)

and also for the SIR model where quarantine is directly introduced on the infected nodes:

Anand et al. Transactions of the Indian National Academy of Engineering 5, 141–148 (2020)

Giordano et al. Nature Medicine 26, 855–860 (2020)

Mancastroppa et al. Phys. Rev. E 102, 020301(R) (2020)

Many other papers have been recently published or submitted.

In these models often several aspects are taken into account which are overlooked in the present paper e.g. hospitalization, asymptomatic individuals, heterogeneous behavior of the individuals. The authors should explain what is the main advance with respect the vast recent literature.

In this respect, the authors consider the simultaneous adoption of three different kinds of measures, however it is not clear in the paper what is the main advantages of introducing this modelization. In particular which are the different effects due to the three strategy of containment? E.g. when you consider the fit of datasets the use of a single strategy (i.e. social distancing) provides a very similar fit of the case where the synergy of the different strategies is considered; also the improvement in the X^2 is very small. So it is not clear what is the advantage of adopting the different strategies, in the description of non-pharmaceutical interventions. In summary the paper introduces a model with a description of three types of non-pharmaceutical interventions, this interventions have been already introduced in different papers even if with some differences. In this context, from a theoretical point of view the authors do not evidence any peculiar or interesting effect due to the presence of these three terms; while a comparison with datasets does not show any advantages in interpretations of the epidemic evolution in the different countries. In this perspective I think that the paper is not suitable for publication without a significant improvement.

Another very important remark is about the model for a generalized lock-down. In your approach only susceptible nodes are isolated in the state P while I expect that a fraction of the whole population in a lock-down is isolated; in particular I expect that the sudden isolation of infected and exposed individuals should have a larger effect than the isolation of susceptible ones. Moreover I think that the process should be described only by the approach in figure 4 i.e. a sudden isolation of a large number of individuals which remain isolated during the whole period of non-pharmaceutical interventions; while other individuals do not self isolates (e.g. they do an essential job), so that after the quick self isolation mu is restored to 0. On the other hand, a self isolation with constant rate, in the whole period of intervention seems to be very unlikely. An even a similar and simpler model of a generalized lock down could be at the beginning of the intervention an instantaneous isolation in the state P of a fraction of individuals (independently of their states). Therefore, I think that the different scenarios and the data fitting should take into account only the approach in figure 4 (clearly with possible different fraction of people that self isolate).

Reviewer #2: REVIEWER COMMENTS TO AUTHORS

Referee report: PONE-D-20-26921

Evaluation

The paper has a number of shortcomings, and would require a much deeper

discussion of several parts (see my main points). Also, the

epidemiological language should be improved in many points. Some cited

papers are weak. Therefore, the authors should (i) do an effort to

(substantially) amend their paper according to the indications reported

below, but especially (ii) put into evidence their findings that are of

main epidemiological interest, particularly in relation to the insights

that the adopted modeling framework would provide in relation to the

understanding of COVID-19 control.

Main points

The proposed SPQEIR model is quite restrictive in its formulation and

therefore the underlying hypotheses should be carefully discussed.

First, P individuals are – according to the stated hypotheses - fully

protected, which corresponds to full segregation forever i.e., during

the entire history of intervention measures. But measures aimed at

confining susceptible people hardly can go beyond home confinement, and

there is strong evidence that – during lockdowns - much COVID

transmission occurred within households (especially during the first

wave). Pairwise, also removed E individuals are fully inactivated in the

SPEIR. Also, this should be discussed carefully. Moreover, as E

individuals can be removed mostly by way of tracing, it is not clear to

me why you do not allow a pairwise option for I individuals (e.g.

asymptomatic or pauci-symptomatic) by the same mechanism. Last, I

understand screening and isolation of actively infected individuals is

incorporated into removal (L99), which is an option. However, did you

handle your removal rate to account for this (perhaps I didn’t note

details)?

L141 “We fit the model to the official number of currently infected

(active) cases, for each considered country.” The authors are surely

aware that published numbers of currently infectious cases poorly

represent true infectious cases. So, this should at least be discussed

more carefully. Moreover, this information risks of being severely

biased when you aim at making comparisons between different countries –

especially during the first wave - because it reflects the inter-country

differences in testing and tracing, making comparisons unreliable.

L>140. Fitting procedures. The authors adopted a nonlinear least squares

procedure citing a rather old textbook whereas the basic statistics of

epidemic data has progressed dramatically in the last twenty years,

first of all maximum likelihood techniques. For example, I do not

understand how the quantity in (3) can be used to document the

improvement in fit compared to the baseline model.

L83 “the time T passed until no new infections occur”, this is quite

wrong at least as far as your model is a deterministic one, as I

understand it is. In the practice of simulation this does not need to be

a problem (and indeed you acknowledge this at a later stage), but the

sentence should be modified.

L119, the effective reproduction number is not explicit in the SIR

model, because the susceptible fraction is not explicit. Therefore, the

formula drawn from ref [26] is an approximation. The problem is that it

is far from being general and rests on a number of hypotheses, which I

find somewhat naive. This also holds for eq (1). I noted that even in

the cited paper [26] the formula is given without a justification. The

formula trivially holds if you assume that the removal by segregation of

the susceptible population occurs rapidly, that is before the

susceptible population is sufficiently depleted. In this case S(t) =

Nexp(-\\�t), so that if you additionally assume that \\�t is small (which

contradicts that segregation occurs fast) and resort to the linear

approximation of the exponential function you arrive to the point.

Anyhow, is this relevant for this paper? On top, I recommend to avoid to

cite whatever paper appeared in this epoch because the quality is not

necessarily good (sometimes poor) and may induce errors in readers.

Legends of simulation exercises are scanty and should be improved.

Other points

> L34, “homogeneous propagation media” is naïve for most readers of the

> Journal; as epidemiologists we speak of “a homogeneously mixing

> population” which is a nowadays somewhat universally agreed

> terminology.

IBM models are not continuous but discrete models (due to their very

structure of simulative models)

L42, "likely" : socially active and at risk of infection

L83 Clearly, the SEIR model by its I curve provides only a very indirect

measure of the pressure on PH system (instead represented by ICU and

hospitals occupancy). Even more so for the economic system. This should

be discussed.

L112 “resulting in the effective reproduction number”, In epidemiology

the effective reproduction number deals with a situation where the

susceptible proportion is depleted below one, as you correctly say in

the subsequent line. Suggest to rephrase.

L126 “We use mean values etc”, please clarify

L129 “with conservation of the total number of individuals, meaning N’ =

0”, the argument goes the other way round, your system fulfills N’=0

implying that N is conserved.

L86 “Mainstream suppression measures against the epidemic aim at

flattening the curve of new infections”, replace “suppression “ by

“mitigation” (flattening the epidemic curve is somewhat different from

suppressing)

L161 “comparative information analysis” this is not an agree terminology

Minor points

I suggest to delete “new” from the abstract and simply state: “an

extended SEIR model including quarantine of susceptible an latently

infected individuals”

L8 “statistical methods allow for accurate characterization of the

population's health state", stated like this is a bit trivial.

L42, "likely" : socially active and at risk of infection

L111 “repression” not appropriate

L117 “physical reduction of a country's population” ?

L182 I suggest to replace “mathematical” in the title with “Simulation”:

it is a simulative analysis

L189 “eradication time” inappropriate wording

L194 Citation 32 is not appropriate. That paper considered a model with

behavioral responses which are not included here.

6. PLOS authors have the option to publish the peer review history of their article (what does this mean?). If published, this will include your full peer review and any attached files.

Reviewer #1: No

Reviewer #2: No

---

## [Author Response · Author response to Decision Letter 0]

29 Jan 2021

Dear editor, dear reviewers,

We thank you for your careful review and for the list of suggested improvements. Please find below detailed responses to the points you raised. In the main text, the parts we edited, following your suggestions, are in red. Please notice that we also added Fig. 7 and updated Fig. 8 and 9 (of the new version) according to what suggested by you and discussed in the answers below. 

Reviewer #1: 

The paper introduces in the SEIR model 3 different types of non-pharmaceutical interventions i.e. social distancing, quarantine of exposed individuals and generalized lock-down. After the outbreak of COVID pandemic several papers have been published in this direction e.g.

Das et al. medRXIV https://doi.org/10.1101/2020.06.04.20122580;

Tomas de-Camino-Beck. medRXIV https://doi.org/10.1101/2020.05.19.20106492;

Perkins et al. Bulletin of Mathematical Biology 82, 118 (2020);

Lai et al. Nature volume 585, 410–413 (2020);

Alrrashed et al Informatics in Medicine Unlocked 20, 100420 (2020)

and also for the SIR model where quarantine is directly introduced on the infected nodes:

Anand et al. Transactions of the Indian National Academy of Engineering 5, 141–148 (2020)

Giordano et al. Nature Medicine 26, 855–860 (2020)

Mancastroppa et al. Phys. Rev. E 102, 020301(R) (2020)

Many other papers have been recently published or submitted.

>We updated out bibliography with recent publications, including some of the ones mentioned above [6, 7, 8, 11, 12, 16, 18, 35, 45, 49, 52, 55, 56, 57]. We also integrated our references with a discussion of relevance of epidemic control not only on the healthcare [17,18,31], but also on the economic system [32,33,34]. Additionally, we discussed the reliability of the empirical data used for model fitting. In fact, we payed particular attention that, across the analysed countries, the share of positive tests was similar [27] and no significant deviation from expected dynamics was observed by studies applying Benford’slaw [28, 29].

Regarding the vast recent modelling literature, several studies have deeply analyzed the effects of specific interventions in selected countries, by statistical methods on simple and interpretable SIR-like models [1, 2, 6, 7, 8, 16, 17, 18, 35, 44], while in our study we consider the plausible impact of a combination of multiple strategies, and the difference from applying single interventions. A similar analysis was done with complex and data-demanding agent-based models [3, 31], which are often less generic, since focusing mostly on one country at a time. Instead, our model provides generic results that can be valid over several countries as shown in Fig. 9. Other works focus on single interventions with stochastic models [45, 55, 59]; we complement them by a different perspective. Many papers are focused on developing future projections [17, 41, 42, 60], but our aim is to discuss how to obtain similar mitigation with different strategies, to inform future decisions. As pointed out by Reviewer#1 below, additional papers account for extra features of importance for pandemic modelling. We discuss this in our response below. We are aware of abundant literature that consider conceptual models and the effect of different parameters (either published [11,12] or as preprints, e.g. Chladnà, medRxiv, 2020). We believe that our additional comparison with empirical data (Fig. 9) can contribute to the discussion about the relevance of such abstract strategies in real-world settings. Finally, our choice of modelling and selection of empirical data is based on socio-political analysis reports (sec 2.2 and Table 1), thus incorporating a multi-disciplinary approach in our study.

In these models often several aspects are taken into account which are overlooked in the present paper e.g. hospitalization, asymptomatic individuals, heterogeneous behavior of the individuals. The authors should explain what is the main advance with respect the vast recent literature.

>Our paper aims at complementing the recent literature by synthetizing several control strategies into a readily interpretable model and by assessing their efficacy both alone and in synergy. This way, we can discuss the merits of each strategy and the possibility of mitigating the curve by a combination of them. The paper focuses on the average effects of such strategies on the infectious curve. We ultimately show that analogous containment levels of the infectious curve can be achieved by alternative synergies of non-pharmaceutical interventions. Hence, policymakers could aim to achieve the desired mitigation by measures targeting different population groups. This synergistic approach still remains of high importance in these second/third waves, where countries need to balance different non-pharmaceutical interventions to keep the infection at bay, while containing the impacts on other aspects of society. Following the remarks by reviewers, and recent literature, we have included isolation of infectious individuals into the model and simulations, on top of isolation of latent carriers. 

We acknowledge that there are aspects of the pandemic found in other studies that were overlooked here, such as hospitalization, asymptomatic individuals, and heterogeneous behaviour of individuals. While these are all important aspects that can be crucial to answer certain questions, they are not needed here to answer the main question of the present paper: how strategies mitigate the infectious curve. Asymptomatic/undetected cases do play a role on the infection curve, but estimating their numbers is affected by large uncertainties. Thus, we opted for a more conceptual model without explicit asymptomatic/undetected compartments. When we compare our framework to empirical data, our goal is mostly to show that the model can reproduce the realistic dynamics observed in real data across a multitude of countries, and that synergies of several measures applied with moderate intensity can still provide the same benefit than a much more intense application of one intervention only. 

On the other hand, to address different questions that require hospitalizations and asymptomatic/undetected cases, our group carried out a distinct study that includes such additional aspects, plus deaths and vaccination, throughout several epidemic waves (https://www.medrxiv.org/content/10.1101/2020.12.31.20249088v1.full).

In this respect, the authors consider the simultaneous adoption of three different kinds of measures, however it is not clear in the paper what is the main advantages of introducing this modelization. In particular which are the different effects due to the three strategy of containment? E.g. when you consider the fit of datasets the use of a single strategy (i.e. social distancing) provides a very similar fit of the case where the synergy of the different strategies is considered; also the improvement in the X^2 is very small. So it is not clear what is the advantage of adopting the different strategies, in the description of non-pharmaceutical interventions. 

>The model links several control strategies to real-world intervention programs, to investigate the possibility of achieving similar mitigation effects by using different strategies. The possibility of choosing among several strategies is of practical importance for decision makers. The impact of said strategies on the infectious curve is investigated in the dedicated sections (3.1.1, 3.1.2, 3.1.3, 3.1.4) and in panels (a) of Fig. 2, 3, 5 and 7). Increasing social distancing delays and decrease the height of the peak of infections. Increasing active protection as well decreases the height of the peak of infections, but it anticipates the occurrence of such peak. Increasing active quarantining also delays and decrease the height of the peak of infections like social distancing, but the same decrease in height of the peak by active quarantining is associated with shorter delay than with social distancing. The effects of measures targeting S and E population groups is also compared with those of isolation of contagious individuals (3.1.4). This shows how much preventive measures can buffer imperfect isolation of I persons. While not described by the model, these strategies are also expected to potentially have different effects on the population at the economic, social and psychological levels. 

In addition, the performed fitting illustrates how well we can explain the evolution of empirical infectious curves with different control strategies. The chi^2 illustrates that the epidemic curves from 6 countries could indeed be fitted equally well (similar X^2, only small changes) by either a strong “social distancing” intervention or by milder \\rho and synergistic measures. This is clearer in Fig.9 of the new version, where we also consider the isolation of infectious individuals. Moreover, we observe a high degree of correlation among some parameters. This is informative about the advantage of adopting a synergy of interventions instead of just one (e.g. only social distancing), in that it achieves a similar mitigation of the infection curve with interventions that can be selected based on the goal of limiting the impact on economy and society. This analysis can thus inform policymakers about the effects of strengthening non-pharmaceutical interventions others than social distancing, as e.g. contact tracing, mass testing, rapid testing.

In summary the paper introduces a model with a description of three types of non-pharmaceutical interventions, this interventions have been already introduced in different papers even if with some differences. In this context, from a theoretical point of view the authors do not evidence any peculiar or interesting effect due to the presence of these three terms; while a comparison with datasets does not show any advantages in interpretations of the epidemic evolution in the different countries. In this perspective I think that the paper is not suitable for publication without a significant improvement.

We complement the analysis of other valuable works, by considering combinations of control interventions in a simple, easily interpretable SIR-like model. Some seminal works did so with Agent-based models (e.g. [31]) but they require large and variegate data sets and are often less generic, since focusing mostly on one country at a time. Here, we highlight a general finding, which we found occurring in multiple countries. Introducing new terms allows for clearer interpretation of the effect of distinct control strategies that can achieve the same mitigation as untargeted, population-wide social distancing. From our experience in the Luxembourg COVID-19 Taskforce, this is useful to balance different options in decision making. The key message of our study is that fostering, strengthening and empowering additional interventions (other than social distancing) can achieve the same mitigation on the infection curve. This is investigated both from a conceptual point of view (Fig. 8) and by considering the evolution of empirical data under mitigation strategies (Fig.9). In the latter, differences in rho might appear small, but in fact correspond to significant changes in people’s lifestyle in terms of measures employed to obtain them (curfew, home office, closure of restaurants, shops, recreational activities, sport clubs and ultimately lockdown). Changes in the second digit of \\rho do correspond to significant changes in lifestyle for a considerable amount of people, see e.g. https://www.medrxiv.org/content/10.1101/2020.12.31.20249088v1.full.

Another very important remark is about the model for a generalized lock-down. In your approach only susceptible nodes are isolated in the state P while I expect that a fraction of the whole population in a lock-down is isolated; in particular I expect that the sudden isolation of infected and exposed individuals should have a larger effect than the isolation of susceptible ones. Moreover I think that the process should be described only by the approach in figure 4 i.e. a sudden isolation of a large number of individuals which remain isolated during the whole period of non-pharmaceutical interventions; while other individuals do not self isolates (e.g. they do an essential job), so that after the quick self isolation mu is restored to 0. On the other hand, a self isolation with constant rate, in the whole period of intervention seems to be very unlikely. An even a similar and simpler model of a generalized lock down could be at the beginning of the intervention an instantaneous isolation in the state P of a fraction of individuals (independently of their states). Therefore, I think that the different scenarios and the data fitting should take into account only the approach in figure 4 (clearly with possible different fraction of people that self isolate).

>The approach described by Reviewer#1 was used for the fitting, in which a fraction of individuals (independently of their state) is isolated or disconnected through the action of all parameters together. Indeed, we used the mu_ld from Fig.4 for the fitting. Following Reviewer#1 comments, we edited the text to make the point clearer (“For the protection parameter, we used μ_ld acting on 4 days (as analysed in Fig. 4) since it better reflects the rapid isolation that happened during the first COVID-19 wave”, Line 369) and we corrected the labels in Fig. 9. 

Reviewer#2: 

Evaluation

The paper has a number of shortcomings, and would require a much deeper discussion of several parts (see my main points). Also, the epidemiological language should be improved in many points. Some cited papers are weak. Therefore, the authors should (i) do an effort to (substantially) amend their paper according to the indications reported below, but especially (ii) put into evidence their findings that are of main epidemiological interest, particularly in relation to the insights that the adopted modeling framework would provide in relation to the understanding of COVID-19 control.

>We thank Reviewer#2 for his/her careful comments and remarks.

Main points

The proposed SPQEIR model is quite restrictive in its formulation and therefore the underlying hypotheses should be carefully discussed. First, P individuals are – according to the stated hypotheses – fully protected, which corresponds to full segregation forever i.e., during the entire history of intervention measures. But measures aimed at confining susceptible people hardly can go beyond home confinement, and there is strong evidence that – during lockdowns - much COVID transmission occurred within households (especially during the first wave). Pairwise, also removed E individuals are fully inactivated in the SPEIR. Also, this should be discussed carefully. 

>We included these points for discussion and disclosed the model assumptions more explicitly (Sec 2.3 and Discussion, lines 473-485). In general, such points would indeed diminish the effectiveness of the control strategies, for which we assess the theoretical impact. We agree that including these aspects would likely make a significant difference on the exact values of the model variables during the epidemic waves. Nevertheless, we believe that they would not impact our take home message, i.e. that different measures impact differently the epidemic curves, but the same desired mitigation of the infectious curve can be obtained by social distancing plus synergies of other measures with potentially smaller socio-economic impact. We indeed proved this for 6 European countries. Fitting their data using either social distancing alone, or a synergy of strategies, results in milder social distancing (higher rho), thus assessing the generality of this finding.

Moreover, as E individuals can be removed mostly by way of tracing, it is not clear to me why you do not allow a pairwise option for I individuals (e.g. asymptomatic or pauci-symptomatic) by the same mechanism. Last, I understand screening and isolation of actively infected individuals is incorporated into removal (L99), which is an option. However, did you handle your removal rate to account for this (perhaps I didn’t note details)?

We agree with Reviewer#2’s remarks. The reason why we originally did not allow a pairwise option for I individuals, i.e. the possibility for I individuals to be removed from the infectious pool by quarantining, is that the first focus of the study was on investigating the role and the importance of preventive measures targeting S and E population groups, like tracing (e.g by app), protection of vulnerable individuals, and social distancing. Nevertheless, we fully agree that these neglect the fact that, once a person has become infectious (I), he/she might be tested and subsequently before having naturally recovered. Comparing the effect of this intervention with preventive ones is therefore very interesting. Hence, we modified the model, simulations and figures to incorporate this advice from reviewer 2. In particular, the ODEs are modified by an extra parameter \\eta handling the removal rate, which in turn modifies R (Eq. 1). Then, we performed its systematic analysis in Sec. 3.1.4 and Fig. 7. The proposed synergies (Sec 3.2 and Fig. 8) now take \\eta into account, as well as the fitting performed in Fig. 9. For fitting, we assumed by default that all countries worked to isolate contagious individuals; hence, the parameter \\eta is associated to all. This is also disclosed in the description of Table 1. In other sections, the text was edited accordingly (Lines 151-154, 434-440). 

L141 “We fit the model to the official number of currently infected (active) cases, for each considered country.” The authors are surely aware that published numbers of currently infectious cases poorly represent true infectious cases. So, this should at least be discussed more carefully. Moreover, this information risks of being severely biased when you aim at making comparisons between different countries – especially during the first wave - because it reflects the inter-country differences in testing and tracing, making comparisons unreliable.

>We agree with the reviewer that published data of currently infected cases do not fully represent the true infectious cases, largely due to a fraction undetected cases, either because asymptomatic or because of lack of sufficient testing. Estimating this fraction of undetected can be based on prevalence studies based on antibodies tests, but ultimately this fraction remains largely uncertain. Thus, we decided not to employ such estimates in our model, as we considered the measured numbers of detected cases sufficient for the scope of our study, which is not to produce projections of the future evolution of the epidemic. Instead, it is intended to show how a synergy of measures could explain the observed curves of infection as nicely as social distancing alone, with lower levels of social distancing itself. 

When choosing the countries for the subsequent analysis, we considered those with similar share of positive tests, aiming at excluding countries with too inefficient testing strategies and thus with overwhelmingly high numbers of undetected cases. We would like to stress that, while we do fit to 6 countries, our aim is not to compare the numbers of active infections between countries, but rather to verify our analysis of the effect of synergies on a number of real-world settings. Moreover, our focus is not on the values of active cases between countries, but rather on their dynamics, i.e. the shape of the infections curves. In addition, recent studies showed no statistically significant divergence of the data of the considered countries from theoretical trends [28,29]. Hence, it is reasonable to assume that these time-series data are capturing plausible time evolutions, which are thus informative of the epidemic dynamic. 

In general, we acknowledge that Reviewer#2’s remark is particularly important for the overall interpretation of the results, so we broadened our discussion and reported the above discussion in the text (Lines 87-95 and 491-498).

L140. Fitting procedures. The authors adopted a nonlinear least squares procedure citing a rather old textbook whereas the basic statistics of epidemic data has progressed dramatically in the last twenty years, first of all maximum likelihood techniques. For example, I do not understand how the quantity in (3) can be used to document the improvement in fit compared to the baseline model.

>We acknowledge that most advanced inference techniques exist and are often employed in other frameworks. For instance, in a recent follow up study that our group developed in https://www.medrxiv.org/content/10.1101/2020.12.31.20249088v1.full, we employed Bayesian inference relying on Markov Chain Monte Carlo Methods to fit an extension of the present model. Here we considered the number of parameters small enough and the problem simple enough to justify the use of a simple non-linear least squares. Thus, the fitting procedure applied in this study relied on a non-linear Python pipeline that is in common use for problems with similar degrees of freedom as ours. To support that the method is enough for the goal, we can observe in Fig. 9 that the parameter values obtained allow to the model simulation to fit reasonably well the time-series data. 

Finally, the reduced chi squared metric of eq.3 was not used for fitting, but to assess the goodness-of-fit, as it is a common diagnostic tool [Maydeu-Olivares, Alberto. "Maximum likelihood estimation of structural equation models for continuous data: Standard errors and goodness of fit." Structural Equation Modeling: A Multidisciplinary Journal 24.3 (2017): 383-394.; but also: Wen, Zhonglin, Kit-Tai Hau, and W. Marsh Herbert. "Structural equation model testing: Cutoff criteria for goodness of fit indices and chi-square test." Acta psychologica sinica 36.02 (2004): 186-194.]. The statistical justifications for this can be found e.g. in [43]. In fact, the chi square is the simplest possible form of the likelihood, obtained when writing the likelihood function assuming that errors on the measurement of the data are Gaussian distributed. In absence of better knowledge on the sources of errors on the available data, Gaussian distributed errors are the most common assumption in mean field models. An intuitive understanding can be gained as follows. The reduced chi square (3) is a quantity based on summing up the square distances of data-points from corresponding simulation values. It does provide a measure of the geometrical distance between the model simulation and the datapoints. Thus, smaller reduced chi square means smaller distance between model simulation and data, i.e. better fit. Thus, computing the quantity (3) for two models (e.g. the baseline model and the model with synergies) and comparing the two obtained numbers allow to asses which of the two models provide a better fit to the data. 

L83 “the time T passed until no new infections occur”, this is quite wrong at least as far as your model is a deterministic one, as I understand it is. In the practice of simulation this does not need to be a problem (and indeed you acknowledge this at a later stage), but the sentence should be modified.

>We agree, thus the sentence has now been modified (Line 108).

L119, the effective reproduction number is not explicit in the SIR model, because the susceptible fraction is not explicit. Therefore, the formula drawn from ref [26] is an approximation. The problem is that it is far from being general and rests on a number of hypotheses, which I find somewhat naive. This also holds for eq (1). I noted that even in the cited paper [26] the formula is given without a justification. The formula trivially holds if you assume that the removal by segregation of the susceptible population occurs rapidly, that is before the susceptible population is sufficiently depleted. In this case S(t) = Nexp(-\\�t), so that if you additionally assume that \\�t is small (which contradicts that segregation occurs fast) and resort to the linear approximation of the exponential function you arrive to the point. Anyhow, is this relevant for this paper? On top, I recommend to avoid to cite whatever paper appeared in this epoch because the quality is not necessarily good (sometimes poor) and may induce errors in readers.

>We agree with Reviewer#2’s analysis and his/her remark that, in fact, an explicit dependence to mu does not hold in case of hard lockdown (e.g Fig. 4) and does not add value to our analysis. Hence, we edited the text accordingly (line), by avoiding citing poorly supported papers and by editing eq. 1 in general terms that are not subject to the mentioned assumptions. The subsequent analysis is performed following the updated eq.1.

As for the remark about recent literature, we agree about being careful when citing non-peer reviewed works from the latest months. We amended our bibliography in this respect.

Legends of simulation exercises are scanty and should be improved.

>We improved the legends in simulation and fitting figures.

Other points

L34, “homogeneous propagation media” is naïve for most readers of the Journal; as epidemiologists we speak of “a homogeneously mixing population” which is a nowadays somewhat universally agreed terminology.

>The text was edited (Line 47).

IBM models are not continuous but discrete models (due to their very structure of simulative models)

>The text was edited (Line 52). 

L42, "likely" : socially active and at risk of infection

>The text was edited (Line 55).

L83 Clearly, the SEIR model by its I curve provides only a very indirect measure of the pressure on PH system (instead represented by ICU and hospitals occupancy). Even more so for the economic system. This should be discussed.

>We edited the text to avoid overstatements about possible direct influences of the I curve on other systems (Lines 106-109), and we discussed and supported its relevance in epidemic management by referring to multidisciplinary studies [17, 18, 31-34]. We did not include such aspects in the current model because they went beyond the question we tackle here. 

L112 “resulting in the effective reproduction number”, In epidemiology the effective reproduction number deals with a situation where the susceptible proportion is depleted below one, as you correctly say in the subsequent line. Suggest to rephrase.

>The text was edited (Lines 141-144).

L126 “We use mean values etc”, please clarify

>The text was edited (Line 155). 

L129 “with conservation of the total number of individuals, meaning N’ = 0”, the argument goes the other way round, your system fulfills N’=0 implying that N is conserved.

>The text was edited (Line 160).

L86 “Mainstream suppression measures against the epidemic aim at flattening the curve of new infections”, replace “suppression “ by “mitigation” (flattening the epidemic curve is somewhat different from suppressing)

>The text was edited (Line 112 and throughout the text).

L161 “comparative information analysis” this is not an agree terminology

>The text was edited (Line 193-196).

Minor points

I suggest to delete “new” from the abstract and simply state: “an extended SEIR model including quarantine of susceptible an latently infected individuals”

>We agree about this and edited the text accordingly.

L8 “statistical methods allow for accurate characterization of the population's health state", stated like this is a bit trivial.

>The text was edited (Line 8).

L111 “repression” not appropriate

>The text was edited (Line 140 and throughout the text). 

L117 “physical reduction of a country's population” ?

>We meant that the active population within a country could diminish after reduced commuters’ activity. For instance, this happened and had a great impact in the country of our institution (Luxembourg). The text was therefore edited (Line 146).

L182 I suggest to replace “mathematical” in the title with “Simulation”: it is a simulative analysis

>The text was edited (Line 214).

L189 “eradication time” inappropriate wording

>The text was edited (Line 221).

L194 Citation 32 is not appropriate. That paper considered a model with behavioral responses which are not included here.

>We agree with Reviewer#1 and we removed the reference not to confuse the reader and edited the text accordingly (Line 224).

---

## [Decision Letter · Decision Letter 1]

23 Feb 2021

PONE-D-20-26921R1

Dynamical SPQEIR model assesses the effectiveness of non-pharmaceutical interventions against COVID-19 epidemic outbreaks

PLOS ONE

Dear Dr. Proverbio,

Thank you for submitting your manuscript to PLOS ONE. After careful consideration, we feel that it has merit but does not fully meet PLOS ONE’s publication criteria as it currently stands. Therefore, we invite you to submit a revised version of the manuscript that addresses the points raised during the review process.

Reviewer #1 has raised some concerns that require an additional revision, in particular regarding the fit and the parameters of the model. 

We look forward to receiving your revised manuscript.

Kind regards,

Michele Tizzoni

Academic Editor

PLOS ONE

Reviewers' comments:

Reviewer's Responses to Questions

**Comments to the Author**

1. If the authors have adequately addressed your comments raised in a previous round of review and you feel that this manuscript is now acceptable for publication, you may indicate that here to bypass the “Comments to the Author” section, enter your conflict of interest statement in the “Confidential to Editor” section, and submit your "Accept" recommendation.

Reviewer #1: (No Response)

Reviewer #2: All comments have been addressed

2. Is the manuscript technically sound, and do the data support the conclusions?

Reviewer #1: Partly

Reviewer #2: Yes

3. Has the statistical analysis been performed appropriately and rigorously? 

Reviewer #1: Yes

Reviewer #2: Yes

4. Have the authors made all data underlying the findings in their manuscript fully available?

Reviewer #1: Yes

Reviewer #2: Yes

5. Is the manuscript presented in an intelligible fashion and written in standard English?

Reviewer #1: Yes

Reviewer #2: Yes

6. Review Comments to the Author

Reviewer #1: The authors now clarify the main claim of their work: "We ultimately show that analogous containment levels of the infectious curve can be achieved by alternative synergies of non-pharmaceutical interventions." So the message of the paper now is clear and one can better evaluate the results. However several important points in my opinion still require improvement.

- The authors claim that similar results can be obtained with different interventions. Figure 8 indeed provides similar curves for the first 4 scenarios where the value of the effective reproductive number is similar, while the evolution of scenarios 5 and 6 is different but these scenarios present a different effective R. In this perspective one could infer that scenarios with similar reproductive number display similar epidemic evolution, which is not surprising. Can you discuss this point? Moreover, you fit the data set using as fitting parameters only social distancing or considering all the parameters of the non-pharmaceutical interventions. Similar quality of the fittings are obtained, evidencing that different approachess can be adopted. However, the fit with the whole parameter set is very similar to the case where only social distancing is present (the parameters rho are close, while mu_ld chi and eta are very small). Therefore one could suppose that social distancing is the main intervention observed in the data. What happens e.g. if you consider in the fit only a lock down, which has been a common intervention in the first wave of COVID 19? You state that important correlations among the fitting parameters is observed. It should be interesting if you are able to show what is the region in the parameter space (rho,mu_ld, eta) where you obtain a nice fitting of the epidemic curves with a similar (small) value of chi^2. In this way one could compare the relevance of the different possible interventions. According to the previous hypothesis this region should correspond to a region at constant value of the reproductive number.

- A small constant value of mu as modelling of protection in my opinion is not realistic. The case where a fraction of the population isolate in a small time well represent a lock down. While a constant rate of isolation during the whole epidemics it seems to me that do not represent a realistic non pharmaceutical intervention. While people should self isolate with a small constant rate? On the other hand, a constant mu could well represent vaccination which is not a pharmaceutical intervention. Moreover, the sentence: "In particular, μ = 0.01 d −1 ..... through isolation, this is unrealistic." seems to be in contrast with the large value of mu of figure 2.

- In the lock down description where a large value of mu is activated for a small time to isolate quickly a finite fraction of the population I do not understand why only susceptible people are put into quarantine. I expect that a general measure as a lock down involves all the population including also exposed and infected individuals. Indeed the aim of a lock down is not only protection of the susceptible nodes but also isolation of potentially infected people.

- 10 days from the onset of the first contagion is an arbitrary choice. The importance of an early adoption of measure of containment is a well known fact. Clearly 10 days belongs to such early adoption framework what happens if measures are taken after a longer time. The different strategies are still equivalent?

- I do not understand the sentence: "However, we notice that values of ρ ' 0.3 or lower are more effective in mitigating the epidemic faster.?" clearly the smaller is rho the more is effective the measure to contain the epidemics, but what is the relevance of this comment?

Reviewer #2: (No Response)

7. PLOS authors have the option to publish the peer review history of their article (what does this mean?). If published, this will include your full peer review and any attached files.

Reviewer #1: No

Reviewer #2: **Yes: **Piero Manfredi

---

## [Author Response · Author response to Decision Letter 1]

2 Apr 2021

Dear editor, dear reviewers,

We are glad that Reviewer #2 is entirely satisfied by our revision, and we thank Reviewer #1 for the list of suggested improvements, which encouraged us to significantly extend our work, and making it more complete. Please find below detailed responses to the points that were raised. Our responses are in blue. In the main text, the parts that we edited based on comments from Reviewer #1 are in red. Following Reviewer #1 suggestions, we performed the analysis of further interesting aspects that complement the main findings reported in the manuscript, and we created a Supplementary Information S1 Text to discuss those additional analysis.

Reviewer #1: The authors now clarify the main claim of their work: "We ultimately show that analogous containment levels of the infectious curve can be achieved by alternative synergies of non-pharmaceutical interventions." So the message of the paper now is clear and one can better evaluate the results. However several important points in my opinion still require improvement.

- The authors claim that similar results can be obtained with different interventions. Figure 8 indeed provides similar curves for the first 4 scenarios where the value of the effective reproductive number is similar, while the evolution of scenarios 5 and 6 is different but these scenarios present a different effective R. In this perspective one could infer that scenarios with similar reproductive number display similar epidemic evolution, which is not surprising. Can you discuss this point? 

The point raised by Reviewer #1 is correct. It is indeed known that epidemic dynamics are driven by the control parameter R_eff. However, there are multiple ways to achieve the same R_eff. Indeed, R_eff results from different combinations of finer-gained parameters. Figure 8 shows how to achieve fine-tuning of different interventions by acting on specific parameters, which could eventually be lumped into the global R_eff as explained in Eq. 1. While Fig. 8 displays a limited number of example scenarios, others can be simulated with our companion Shinyapp (https://jose-ameijeiras.shinyapps.io/SPQEIR_model/). Following Rev. #1’s suggestion, we discussed this point further in the text (Sec. 3.2). 

- Moreover, you fit the data set using as fitting parameters only social distancing or considering all the parameters of the non-pharmaceutical interventions. Similar quality of the fittings are obtained, evidencing that different approachess can be adopted. However, the fit with the whole parameter set is very similar to the case where only social distancing is present (the parameters rho are close, while mu_ld chi and eta are very small). Therefore one could suppose that social distancing is the main intervention observed in the data. 

The fitting called “only social distancing” is primarily performed to assess a reasonable value for R_eff that is less sensitive to the identifiability issue (fitting degenerate parameters) discussed in the text. This is done to make a comparison with the R_eff value obtained with fine-grained parameters fitting. Indeed, we observe that social distancing plays an important role even for the other combinations of interventions, which is not surprising as it is the main population-wide intervention during European-like lockdowns (as also reported in Table 1), while others are more targeted to individuals. Nonetheless, other parameters, despite being low, induce non-negligible impacts (for instance, a moderately high number of individuals flow into the outer compartments, as reported in Table1 of SI Text). We discuss this impact in the main text (Sec. 3.3 and 4) as well as in the new Supplementary Information SI Text (SI Sec. 2), which was added for this purpose. 

- What happens e.g. if you consider in the fit only a lock down, which has been a common intervention in the first wave of COVID 19?

Our strategy was based on fitting only those parameters that are associated with interventions observed and reported in the considered countries (as discussed in Table 1 of Main Text). Hence, we did not perform different fitting with alternative parameters combinations, that were instead not observed in reality (cf. ACAPS database reference in Main Text). As this goes beyond the scopes of the present manuscript, further studies might investigate these aspects. To improve the discussion, we expanded this point in the text (Sec. 3).

- You state that important correlations among the fitting parameters is observed. It should be interesting if you are able to show what is the region in the parameter space (rho,mu_ld, eta) where you obtain a nice fitting of the epidemic curves with a similar (small) value of chi^2. In this way one could compare the relevance of the different possible interventions. According to the previous hypothesis this region should correspond to a region at constant value of the reproductive number.

The identifiability of epidemiological parameters is indeed an interesting point. To better explore the parameter space, we expanded our previous analysis, that was based on a gradient descent non-linear fitting from the lmfit python library. The new extension of our analysis was performed with a Bayesian Inference framework based on Markov Chain Monte Carlo (MCMC) methods. This allowed a better exploration of the parameters space, highlighting regions corresponding to similar R_eff values and similar fitness to data. This additional analysis is now included in the new Supplementary Information SI Text (SI Sec. 2). Figs. 5-7 outline the areas of parameter space (for different couples of parameters and different countries) corresponding to a high posterior probability, and thus a good fitness of the model to the data.

- A small constant value of mu as modelling of protection in my opinion is not realistic. The case where a fraction of the population isolate in a small time well represent a lock down. While a constant rate of isolation during the whole epidemics it seems to me that do not represent a realistic non pharmaceutical intervention. While people should self isolate with a small constant rate? On the other hand, a constant mu could well represent vaccination which is not a pharmaceutical intervention. Moreover, the sentence: "In particular, μ = 0.01 d −1 ..... through isolation, this is unrealistic." seems to be in contrast with the large value of mu of figure 2.

We agree with Reviewer #1’s comment and we edited the considered sentences accordingly. 

In general, small \\mu values are investigated in their corresponding section to complete the conceptual analysis, but only high values during a short time were used to perform the country-wise fitting, thus mimicking the beginning of a lockdown in Europe and reproducing the behavior discussed in Fig. 4.

- In the lock down description where a large value of mu is activated for a small time to isolate quickly a finite fraction of the population I do not understand why only susceptible people are put into quarantine. I expect that a general measure as a lock down involves all the population including also exposed and infected individuals. Indeed the aim of a lock down is not only protection of the susceptible nodes but also isolation of potentially infected people.

This is indeed an important point. From a modelling perspective, all outfluxes from the main epidemic compartments (framed in Fig.1) to Q and P compartments correspond to isolating measures. During the whole intervention period, all outflux parameters are active, thus modelling the observed interventions that were applied during European-like lockdowns, which also targeted potentially infected people as explained in Table 1. 

On the other hand, we named the intervention parameters after the compartment they act upon. So the action of “protection” is on susceptibles only, whereas the removal of individuals from the infectious compartments (framed in Fig.1) comes from the combination of all outflux parameters.

- 10 days from the onset of the first contagion is an arbitrary choice. The importance of an early adoption of measure of containment is a well known fact. Clearly 10 days belongs to such early adoption framework what happens if measures are taken after a longer time. The different strategies are still equivalent?

We agree with the reviewer that 10 days is to some extent arbitrary. This aspect is indeed of interest for readers, as the delay \\tau represents an extra parameter that might influence the dynamics. To answer in detail and to improve the content of our manuscript, we performed additional simulations and included an extended discussion section in the Supplementary Information SI Text (SI Sec. 1). We observe that the qualitative evolution (trends, dependence on parameters, etc.) of the epidemic curve under interventions does not change, and its quantitative details (height of the peak, absolute number of days until mitigation) change according to Figs. SI 1-4. Of particular interest is the resulting tradeoff of prompt-but-weak and delayed-but-strong interventions, which cause different quantitative outcomes. Interestingly, such outcomes are currently observed in several European countries. 

- I do not understand the sentence: "However, we notice that values of ρ ' 0.3 or lower are more effective in mitigating the epidemic faster.?" clearly the smaller is rho the more is effective the measure to contain the epidemics, but what is the relevance of this comment?

In Fig.2c of main text, we highlight a non-monotonous and, more intriguing, non-linear relationship between the peak height (and the mitigation timing) and the intervention parameters. The aforementioned sentence was intended to stress this point, and the fact that there are certain parameter values that minimize the peak and the timing. However, we acknowledge that it was initially not entirely clear, and we now edited the text accordingly. 

We thank Reviewer #1 for his/her feedback, which led us to perform further analysis and increase the depth of the manuscript.

---

## [Decision Letter · Decision Letter 2]

16 Apr 2021

PONE-D-20-26921R2

Dynamical SPQEIR model assesses the effectiveness of non-pharmaceutical interventions against COVID-19 epidemic outbreaks

PLOS ONE

Dear Dr. Proverbio,

Thank you for submitting your manuscript to PLOS ONE. After careful consideration, we feel that it has merit but does not fully meet PLOS ONE’s publication criteria as it currently stands. Therefore, we invite you to submit a revised version of the manuscript that addresses the points raised during the review process.

Please, in your revision address the minor issues raised by Reviewer 1. 

We look forward to receiving your revised manuscript.

Kind regards,

Michele Tizzoni

Academic Editor

PLOS ONE

Journal Requirements:

Reviewers' comments:

Reviewer's Responses to Questions

**Comments to the Author**

1. If the authors have adequately addressed your comments raised in a previous round of review and you feel that this manuscript is now acceptable for publication, you may indicate that here to bypass the “Comments to the Author” section, enter your conflict of interest statement in the “Confidential to Editor” section, and submit your "Accept" recommendation.

Reviewer #1: (No Response)

2. Is the manuscript technically sound, and do the data support the conclusions?

Reviewer #1: Yes

3. Has the statistical analysis been performed appropriately and rigorously? 

Reviewer #1: Yes

4. Have the authors made all data underlying the findings in their manuscript fully available?

Reviewer #1: Yes

5. Is the manuscript presented in an intelligible fashion and written in standard English?

Reviewer #1: Yes

6. Review Comments to the Author

Reviewer #1: The authors have answered to my previous questions and in general the paper is improved accordingly.

However some points still require an improvement:

My comment on the fact that during a lock down not only susceptible individuals are removed but

also infected and exposed, has not been addressed clearly:

"This is indeed an important point. From a modelling perspective, ........the

combination of all out-flux parameters"

If this means that during the short period of isolation of individuals (4 days) you set mu=mu_ld

but also eta=mu_ld and xi=mu_ld so that also exposed and infected individuals are removed with the same rate?

In my opinion this should be a correct approach: during a lock down the same fraction of S, E and I individuals are typically protected. If this is the case it should be explained, otherwise if you set only mu=mu_ld, the model isolate only susceptible individuals during a lock down which is not very likely and some explanations should be added.

The authors introduce, in the supplementary, the analysis of the posterior probability distribution for different values of the parameters, evidencing that a nice fit of the experimental data is obtained in a wide region of the parameter set. This nice point answer to one of my previous question. However, I do not understand in figures 5,6 and 7 when you plot the probability as a function of two of the parameters how do you fix the value of the other parameters which are not consider in the plot.

There is a typo at page 12: "Sec. ??"

7. PLOS authors have the option to publish the peer review history of their article (what does this mean?). If published, this will include your full peer review and any attached files.

Reviewer #1: No

---

## [Author Response · Author response to Decision Letter 2]

3 May 2021

Dear editor, dear reviewers,

Once again, we thank the reviewers for their further suggestions and improvements. Please find below detailed answers to the questions raised, and explanations on how we tackled the points specified by reviewer #1. Our responses are in blue. In the main text, the parts we edited following your suggestions are in red. 

- Reviewer #1: The authors have answered to my previous questions and in general the paper is improved accordingly.

However some points still require an improvement:

- My comment on the fact that during a lock down not only susceptible individuals are removed but also infected and exposed, has not been addressed clearly:

"This is indeed an important point. From a modelling perspective, ........the

combination of all out-flux parameters" If this means that during the short period of isolation of individuals (4 days) you set mu=mu_ld but also eta=mu_ld and xi=mu_ld so that also exposed and infected individuals are removed with the same rate?

In my opinion this should be a correct approach: during a lock down the same fraction of S, E and I individuals are typically protected. If this is the case it should be explained, otherwise if you set only mu=mu_ld, the model isolate only susceptible individuals during a lock down which is not very likely and some explanations should be added.

We fully agree with Reviewer #1 that the suggested fitting scheme is more realistic. Hence, following the reviewer’s suggestion, we have now tested both methods (mu=mu_ld, as we employed so far, and all parameters=mu_ld during the initial 4 days of rapid isolation). While the latter is more realistic, as mentioned, the “mu=mu_ld” method is better connected to our conceptual analysis. Thus, testing both methods give an indication if a more complex model improves the goodness of fit.

Hence, we performed the fitting to country data with one method at a time, thus obtaining two sets of fitted parameters rho, mu_ld, eta etc., like in Fig.9. Our results confirm that, during the first epidemic wave, the two approaches yield similar quantitative results, as the fitted parameters differ only in the second significative digit while the reduced chi-squared metrics gets slightly worse. Thus, we can conclude the two approaches reproduce the epidemic dynamics equally well, leaving our conclusions unaltered. For illustration, we report fitting values for some example countries. Others display similar subtle changes.

Denmark

Method; rho; mu_ld; chi’; eta; R; chi2red

mu_ld acting on S; 0.35; 0.003; 0; 0.0014; 0.73; 3.8864e-09

mu_ld acting on all compartments; 0.36; 0.019; 0; 0.0013; 0.73; 6.2272e-09

Israel

Method; rho; mu_ld; chi’; eta; R; chi2red

mu_ld acting on S; 0.31; 0.001; 0.001; 0.004; 0.7; 1.6054e-08

mu_ld acting on all compartments; 0.30; 0.001; 0.001; 0.001; 0.7; 4.4888e-08

Switzerland

Method; rho; mu_ld; chi’; eta; R; chi2red

mu_ld acting on S; 0.28; 0.001; 0; 0.001; 0.67; 6.0977e-09

mu_ld acting on all compartments; 0.276; 0.001; 0; 0.001; 0.66; 6.9493e-09

This is due to the fact that, at the beginning of the pandemic, the S compartment contains by far the greatest number of individuals (since S~N); hence, parameters related to the S compartment have a larger impact on the infection dynamics. In addition, the overall impact of mu_ld is smaller than that of other parameters, such as rho, because of the nature of European-like lockdowns. In fact, the variable P in our model, controlled by the parameter mu_ld, represents the compartment of individuals with zero probability of getting infected, which correspond to a minority of people, e.g. to elderly people fully isolated and not the majority of individuals who had still some external contacts due to e.g. going to work, having a family member going to work, doing the groceries etc. The effect of the lockdown on these individuals is represented in our model by the parameter rho, which describes a decreased (but not an absence) level of social interactions.

Following Reviewer#1, we edited the text (lines 386-389) to make this point clearer. 

- The authors introduce, in the supplementary, the analysis of the posterior probability distribution for different values of the parameters, evidencing that a nice fit of the experimental data is obtained in a wide region of the parameter set. This nice point answer to one of my previous question. However, I do not understand in figures 5,6 and 7 when you plot the probability as a function of two of the parameters how do you fix the value of the other parameters which are not consider in the plot.

We now realised this part was not clearly explained in the text. An important point here is that the parameters are not statistically independent. The posterior distribution for a particular parameter depends on the values of other parameters. Hence, to visualise a joint posterior distribution of 2 parameters from a high dimensional one, we need to first marginalise it against the other parameters. In probability, marginalising corresponds to projecting high dimensional distributions to the dimensions of interest by integrating over the remaining (“marginalised-out”) parameters. As an example, taken from https://en.wikipedia.org/wiki/Marginal_distribution, to obtain the marginal probability of x from the joint probability distribution of x and y we would take

p_X (x)= ∫_y▒〖p_(X|Y) (x|y) p_Y (y)dy〗

This can also be exemplified by the following figure (CC0, taken from https://en.wikipedia.org/wiki/Marginal_distribution), where a 2D distribution (highlighted by the green ellipse) is marginalised over X (leading to the 1D red p(y) distribution) and Y (leading to the 1D blue p(x) one), respectively. 

[figure shown in attached .docx fie]

Similarly, we marginalised the full distribution obtained by MCMC chains to produce Figures 5, 6 and 7 of Supplementary Information. We added the word marginal to clarify this point, and discussed it further in Sup Mat (end of Sectioon 2.2. of Supplementary Information).

Finally, it has been reported (cf. Table 1) that some countries did not employ certain strategies. Thus, for those countries, the corresponding parameters were fixed to default values as described in Sec. “The extended SPQEIR model to reflect mitigation strategies”.

- There is a typo at page 12: "Sec. ??"

The typo has now been corrected, we thank Reviewer#1 for noticing.

---

## [Decision Letter · Decision Letter 3]

10 May 2021

Dynamical SPQEIR model assesses the effectiveness of non-pharmaceutical interventions against COVID-19 epidemic outbreaks

PONE-D-20-26921R3

Dear Dr. Proverbio,

We’re pleased to inform you that your manuscript has been judged scientifically suitable for publication and will be formally accepted for publication once it meets all outstanding technical requirements.

Kind regards,

Michele Tizzoni

Academic Editor

PLOS ONE

Additional Editor Comments (optional):

Reviewers' comments:

Reviewer's Responses to Questions

**Comments to the Author**

1. If the authors have adequately addressed your comments raised in a previous round of review and you feel that this manuscript is now acceptable for publication, you may indicate that here to bypass the “Comments to the Author” section, enter your conflict of interest statement in the “Confidential to Editor” section, and submit your "Accept" recommendation.

Reviewer #1: All comments have been addressed

2. Is the manuscript technically sound, and do the data support the conclusions?

Reviewer #1: (No Response)

3. Has the statistical analysis been performed appropriately and rigorously? 

Reviewer #1: Yes

4. Have the authors made all data underlying the findings in their manuscript fully available?

Reviewer #1: Yes

5. Is the manuscript presented in an intelligible fashion and written in standard English?

Reviewer #1: Yes

6. Review Comments to the Author

Reviewer #1: The authors have answered to all my previous comments and now the article can be published on Plos One

7. PLOS authors have the option to publish the peer review history of their article (what does this mean?). If published, this will include your full peer review and any attached files.

Reviewer #1: No

---

## [Editor Report · Acceptance letter]

12 May 2021

PONE-D-20-26921R3 

Dynamical SPQEIR model assesses the effectiveness of non-pharmaceutical interventions against COVID-19 epidemic outbreaks  

Dear Dr. Proverbio:

I'm pleased to inform you that your manuscript has been deemed suitable for publication in PLOS ONE. Congratulations! Your manuscript is now with our production department. 

Kind regards, 

on behalf of

Dr. Michele Tizzoni 

Academic Editor

PLOS ONE